# SPAR: Support-Preserving Action Rectification

Jiaxin Zhao [1]   Weihang Pan [1]   Xun Liang [1]   Binbin Lin [1]

## Abstract

Offline policy improvement faces an inherent conflict between maximizing value and fitting the data distribution. While in-sample weighted regression is stable, it suffers from over-conservatism that suppresses high-value actions in the distribution tail; conversely, gradient-based approaches often exhibit a fitting-optimization conflict of gradients, which drive the policy off the data manifold. To address this, we propose **S**upport-**P**reserving **A**ction **R**ectification (SPAR), which reframes global learning as a local residual rectification anchored to a frozen pure behavior cloning policy. This framework performs fine-grained fitting and local policy improvement in the residual space, thereby contracting the search space. We further introduce Latent Self-Imitation, utilizing a latent-sampling weighted-regression mechanism to address fitting-improvement gradient conflict in the residual space. Theoretically, we prove this mechanism eliminates the manifold-normal drift of standard value gradients, while extensive D4RL experiments show SPAR extracts significant gains from suboptimal baselines to achieve state-of-the-art performance.

## 1. Introduction

Offline reinforcement learning (offline RL) aims to learn a decision-making policy purely from a fixed dataset of logged interactions without any additional environment access (Fu et al., 2020; Levine et al., 2020; Prudencio et al., 2023). A central limitation is finite coverage: the dataset only supports a non-uniform subset of the state–action space, leaving large regions effectively unconstrained by data (Fujimoto et al., 2019; Kumar et al., 2019; Wu et al., 2019). To outperform suboptimal policy, algorithms must perform counterfactual queries—exploring action combinations rarely

[1]Zhejiang University. Correspondence to: Weihang Pan <panweihang@zju.edu.cn>.

*Proceedings of the 43$^{rd}$ International Conference on Machine Learning*, Seoul, South Korea. PMLR 306, 2026. Copyright 2026 by the author(s).

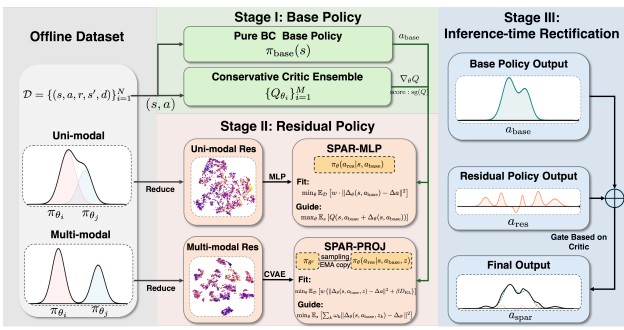

*Figure 1.* An illustrative diagram of the Support-Preserving Action Rectification (SPAR) method.

seen in the dataset to discover potential gains (Levine et al., 2020; Kostrikov et al., 2021). As a result, reliable policy improvement in offline RL must reconcile two conflicting objectives: leveraging value estimates to discover potential gains while strictly respecting the dataset support to prevent out-of-distribution (OOD) collapse at deployment (Wu et al., 2022; Nakamoto et al., 2023).

Existing policy improvement methods broadly fall into two paradigms, each exhibiting a distinct geometric failure mode under limited coverage. In-support (in-sample) learning treats improvement as weighted regression toward actions present in the dataset (Peng et al., 2019; Nair et al., 2020; Kostrikov et al., 2021). While this approach is robust to OOD querying, its maximum-likelihood bias emphasizes high-density regions of the behavior distribution, which can systematically suppress rare-but-high-value actions that lie in the long tail (Florence et al., 2022). Geometrically, overfitting discrete support shatters the continuous action manifold, obstructing meaningful interpolation and exploration along tangential directions and trapping the policy in suboptimal regions (Wang et al., 2022; Kang et al., 2023; Chen et al., 2023). In contrast, value-guided optimization updates the policy by ascending the gradient of a learned critic (Kumar et al., 2020; Fujimoto & Gu, 2021; Wang et al., 2022; Chen et al., 2024). However, in the offline regime the critic must extrapolate outside the data neighborhood, where approximation artifacts can create sharp spurious peaks that dominate the gradient field, causing the policy to exploit critic errors rather than true returns. Value gradients therefore contain large normal components that push actions off the data manifold into infeasible adversarial regions, destabilizing the policy and leading to catastrophic

forgetting (Kirkpatrick et al., 2017; Fujimoto et al., 2019; Sinha et al., 2022; Roderick et al., 2023; Geng et al., 2024).

In summary, the limitations of existing paradigms can be distilled into two levels: the structural coupling of conflicting objectives on a single policy via gradient backpropagation, and the practical challenge of balancing these objectives during optimization. Motivated by these issues, we use an objective-decoupled design that assigns fitting and improvement to separate stages with clear priorities. By implementing these objectives with varying intensities across separate networks and subsequently fusing them, we can prevent catastrophic forgetting (Silver et al., 2018; Li et al., 2023). Concurrently, it is crucial to identify an improvement method that balances the fitting objective, ensuring the policy remains on-manifold while retaining sufficient exploration.

We propose **Support-preserving Action Rectification (SPAR)**, as shown in Fig. 1. Leveraging the spectral bias of neural networks (Rahaman et al., 2019), we train a pure behavior cloning(BC) base policy $\pi_{\text{base}}$ as a structural backbone that captures the global manifold topology. Building on residual policy learning, we parameterize the learned policy as $\pi_\theta(s) = \pi_{\text{base}}(s) + \Delta_\theta(s, \pi_{\text{base}}(s))$, where $\pi_{\text{base}}(s)$ is a base anchor and $\Delta_\theta$ is a residual policy capturing local corrections. This decomposition converts unconstrained optimization over the full action space into a localized refinement problem around a data-consistent scaffold (residual space), thereby shrinking the effective search space and preserving manifold-aware constraints from $\pi_{\text{base}}$. Second, SPAR introduces a latent self-imitation procedure inspired by self-imitation learning (Oh et al., 2018): we sample residual candidates generated by the residual policy and combine them with base actions, then weight these synthesized actions by conservative value estimates, and finally regress $\Delta_\theta$ toward this value-weighted empirical residual distribution (Peng et al., 2019; Kostrikov et al., 2021). This procedure retains controlled exploration while rigorously restricting update directions to value-weighted residual support, addressing the geometric normal-drift pathology of unconstrained value gradients, balancing the two objectives. The three-stage design operationalizes this decoupling by separating behavior fitting, conservative value estimation, and local residual improvement, so that each stage has an isolated and interpretable role. In practice, SPAR-PROJ can be used as a unified default for multimodal or uncertain residual geometry, while SPAR-MLP is a lightweight specialization for compact unimodal residuals.

Our contributions span theoretical insight, algorithm design, and empirical validation:

- We theoretically quantify the reduction in the policy search space achieved by residual learning and the gradient conflicts between critic guidance and fitting objectives.

- We propose **SPAR** (Support-Preserving Action Rectification), a framework that decouples fitting and improvement objectives. SPAR fits fine-grained residual details by analyzing the action residual distribution and improves the policy locally via latent self-imitation.
- We evaluate SPAR on the D4RL benchmark and observe strong performance across diverse domains and dataset qualities.

**Conflict of Interest Disclosure.** The authors declare no financial conflicts of interest.

## 2. Related Work

### 2.1. Offline Policy Improvement via In-Support Learning

A prevalent paradigm restricts policy updates to the dataset support, typically casting improvement as weighted supervised regression. Classic methods like AWR, AWAC, and IQL extract policies via advantage-weighted behavioral cloning, avoiding explicit OOD queries to ensure stability (Peng et al., 2019; Nair et al., 2020; Kostrikov et al., 2021; Wang et al., 2020). Another line of work performs in-sample backups or evaluation to avoid querying unseen actions (Zhang et al., 2023; Xiao et al., 2023). Recent works extend this to diffusion or flow models, leveraging their expressivity to capture multi-modal distributions (Wang et al., 2022; Kang et al., 2023; Chen et al., 2023; Park et al., 2025; Zhang et al., 2025). While safe, their maximum-likelihood bias towards high-density regions tends to cause discretization of the fitted manifold. Drawing inspiration from Self-Imitation Learning (SIL), SPAR expands the sample source via sampling in latent space to perform in-domain interpolation, thereby better exploiting these potential gains. Unlike global diffusion or flow policies, SPAR performs this sampling in the local residual space around a frozen BC anchor.

### 2.2. Offline Policy Improvement via Gradient Guidance

A complementary paradigm seeks improvement by directly maximizing learned value estimates, typically manifesting as gradient-based policy optimization (Kumar et al., 2020; Cheng et al., 2022) or value-gradient guided generation (Wang et al., 2022; Kang et al., 2023). Recent diffusion- and flow-style methods refine this idea through constrained diffusion actor updates, flow Q-learning, flow actor-critic updates, and one-step flow policy extraction (Fang et al., 2025; Park et al., 2025; Chae et al., 2026; Nguyen & Yoo, 2026). Despite the inclusion of pessimistic regularization, trust regions, or projection constraints for safety (Chen et al., 2024), the non-smooth nature of neural networks within the sample neighborhood can still yield spurious high-value artifacts, typically manifesting as sharp peaks with high curvature

that obscure genuine smooth high-value interpolations on the manifold. In this scenario, the optimization process against the critic gradient essentially degenerates into generating adversarial examples for the critic (Sinha et al., 2022). Consequently, the guidance signals contain large normal components relative to the plausible distribution manifold, pushing the policy into physically infeasible regions. In contrast, SPAR employs a derivative-free sampling mechanism in the anchored residual manifold, avoiding explicit gradient ascent on the action generator.

### 2.3. Residual Reinforcement Learning (Residual RL)

Residual RL parameterizes the policy as a correction to a baseline, which is designed to facilitate safe incremental adaptation in robotics (Silver et al., 2018; Johannink et al., 2019). In the offline setting, this architecture provides a natural regularization by anchoring the optimization to the dataset manifold via the baseline behavior (Li et al., 2023). However, standard residual parameterization does not strictly prevent value-guided updates from drifting off-support. SPAR addresses this by reformulating the residual update as a weighted regression on high-value samples, thereby structurally confining the policy correction to the empirical support.

## 3. Method

We propose **SPAR** (Support-Preserving Action Rectification), a three-stage pipeline for baseline-anchored offline improvement. **Stage I** establishes a pure behavior cloning (BC) policy as $\pi_{\text{base}}$ the anchor and a conservative critic ensemble. **Stage II** learns a support-preserving residual policy $\pi_{\text{res}}$. **Stage III** (test-time) performs value-gated rectification, conditionally applying the residual only when robust improvement is predicted.

### 3.1. Stage I: Conservative Anchor

Stage I establishes a stable initialization comprising a base policy and a robust value estimator. We obtain the base policy $\pi_{\text{base}}$ via BC on $\mathcal{D}$, deliberately avoiding advantage-weighted objectives. The base policy provides a stable support anchor; multimodal local corrections are delegated to the residual policy in Stage II.

For value estimation, we employ an ensemble of critics $\{Q_{\theta_i}\}_{i=1}^{M}$ and a state-value network $V_\psi$. To eliminate the overestimation bias typical of OOD bootstrapping, we train these networks using strictly in-sample expectile regression. The default setting uses $\tau = 0.5$, which approximates the conditional mean and keeps $V_\psi(s)$ tied to the behavior policy value. For sparse-reward AntMaze tasks, we use $\tau = 0.9$ to improve critic discriminability; this only affects Stage I value estimation and leaves the shared Stage III gate

unchanged.

We encapsulate the critic uncertainty via a Lower Confidence Bound (LCB)(An et al., 2021):

$$Q_{\text{rob}}(s, a) = \mu_Q(s, a) - \lambda_u \sigma_Q(s, a), \tag{1}$$

where $\mu_Q(s, a)$ is the ensemble mean, $\sigma_Q(s, a)$ is the ensemble standard deviation. This $Q_{\text{rob}}$ serves as a conservative score for Stage II. Once trained, both $\pi_{\text{base}}$ and $Q_{\text{rob}}$ are frozen.

### 3.2. Stage II: Support-Preserving Residual Learning

Stage II parameterizes the policy as a correction around the baseline: $a_{\text{spar}}(s, a) = \pi_{\text{base}}(s) + \text{G}(\pi_{\text{res}}(s, \pi_{\text{base}}(s)))$, where $\pi_{\text{base}}$ is the anchor action output and $\text{G}(\cdot)$ is the gated residual output. The total objective of SPAR is:

$$\mathcal{L}_{\text{total}}(\theta) = \mathcal{L}_{\text{fit}}(\theta) + \lambda_g \mathcal{L}_{\text{guide}}(\theta). \tag{2}$$

Implementation-wise, we introduce a unified weighting component in Sec 3.2.2 that serves the entire residual policy learning framework, applying to both the fitting and latent self-imitation objectives. We then detail the specific parameterization of the residual policy in Sec 3.2.3 and the latent self-imitation mechanism in Sec 3.2.4.

#### 3.2.1. SPACE CONTRACTION IN RESIDUAL LEARNING

Residual learning achieves efficiency by contracting the hypothesis space volume. Let $\mathcal{A} \subset \mathbb{R}^d$ denote the global action space reduced by the static dataset with diameter $D_{\mathcal{A}} = \sup_{x,y \in \mathcal{A}} \|x - y\|_2$, and define the global search space as $\Omega_{\text{global}} = \mathcal{A}$. For residual learning, we restrict refinement to a local neighborhood of a frozen base policy $\pi_{\text{base}}$. The locality scale is quantified by the $(1-\rho)$-quantile of residual magnitudes measured directly from the offline dataset:

$$\delta_\rho \triangleq \inf \left\{ \delta > 0 : \Pr_{(s,a) \sim \mathcal{D}} \left( \|a - \pi_{\text{base}}(s)\|_2 \leq \delta \right) \geq 1 - \rho \right\}, \tag{3}$$

which defines the residual search space $\Omega_{\text{res}} = \{\Delta a \mid \|\Delta a\|_2 \leq \delta_\rho\}$ with effective diameter $D_{\text{res}} = 2\delta_\rho$.

Under mild regularity assumptions ($Q(s, \cdot)$ is $L$-Lipschitz continuous and value observations are $\sigma$-sub-Gaussian), the effective data requirement for identifying an $\epsilon$-optimal action scales with the metric entropy of the search region. By restricting the search to a local neighborhood around the base policy, the hypothesis space volume is contracted, yielding a polynomial-logarithmic reduction in the covering number, thereby concentrating the available sample budget to facilitate reliable estimation at resolution $\epsilon/(2L)$. We formally quantify this statistical benefit in the following theorem (the proof is in A.1):

**Theorem 3.1.** *Let $\epsilon > 0$ denote the optimality gap tolerance* $(Q(s, \pi(s)) \geq \max_{a \in \Omega} Q(s, a) - \epsilon)$, $\beta \in (0, 1)$ *the probability that the returned action is not $\epsilon$-optimal within $\Omega$, and $\sigma > 0$ the sub-Gaussian scale parameter of value observations. Under the assumptions that $Q(s, \cdot)$ is L-Lipschitz continuous and value estimates are $\sigma$-sub-Gaussian, the number of samples required to identify an $\epsilon$-optimal action within search region $\Omega$ with probability at least $1 - \beta$ satisfies*

$$N(\epsilon, \Omega) = \tilde{\Theta}\left(\frac{\sigma^2}{\epsilon^2} \cdot \mathcal{N}(\Omega, \frac{\epsilon}{2L}) \cdot \frac{d}{\beta} \cdot \log D\right), \quad (4)$$

*where $\tilde{\Theta}$ absorbs linear factors, $\mathcal{N}(\Omega, r) \propto D^d$ and $d$ is the dimension of action space.*

Since $\delta_\rho < D_{\mathcal{A}}$ is observed in the ten selected D4RL environments as shown in Fig. 4 in Appendix A, this contraction yields a polynomial-logarithmic reduction in estimation complexity. The trade-off is quantified by the approximation error

$$\varepsilon_{\text{app}}(s; \delta_\rho) = \max_{a \in \mathcal{A}} Q(s, a) - \max_{\|a - \pi_{\text{base}}(s)\| \leq \delta_\rho} Q(s, a), \quad (5)$$

which decomposes into the data coverage gap $\varepsilon_{\text{cov}}$ and localization bias $\varepsilon_{\text{loc}}$. The latter admits the following bound (the proof is in A.2):

**Lemma 3.2.** *Let $a^\mu(s)$ denote the optimal action within the behavior support $\text{supp}(\mu) = \{a \mid \mu(a|s) > 0\}$. Under the L-Lipschitz continuity of $Q(s, \cdot)$,*

$$\varepsilon_{\text{loc}}(s; \delta_\rho) \leq L \cdot \left[\|a^\mu(s) - \pi_{\text{base}}(s)\|_2 - \delta_\rho\right]_+, \quad (6)$$

*where $[x]_+ \triangleq \max(x, 0)$ denotes the positive part operator.*

By construction, the $\delta_\rho$-neighborhood preserves $1 - \rho$ fraction of the behavior support mass, ensuring $\|a^\mu(s) - \pi_{\text{base}}(s)\|_2 \leq \delta_\rho$ for most states. Consequently $\varepsilon_{\text{loc}}$ remains modest in practice. Crucially, SPAR trades this controlled bias for huge gains in statistical efficiency, converting an ill-posed global optimization problem into a well-posed local refinement task while respecting the fundamental boundary imposed by data coverage $\varepsilon_{\text{cov}}$.

### 3.2.2. ADAPTIVE WEIGHTING: FILTERING AND SENSITIVITY

For $\mathcal{L}_{\text{guide}}$ implemented by latent self-imitation and for all $\mathcal{L}_{\text{fit}}$ implementations, we adopt a unified weighting pattern $w(s, a)$ based on the normalized advantage $\tilde{A}(s, a) = (Q_{\text{rob}}(s, a) - Q_{\text{rob}}(s, a_{\text{base}}))/\sigma_Q$ to perform weighted regression. We structure the weighting along two orthogonal axes, yielding four distinct regimes:

- **Sensitivity (Uniform vs. Weighting)** Controls positive-gain scaling. Exponential weighting defines the base magnitude $w_{\text{base}} = \exp(\tilde{A}/T)$ to approximate Boltzmann

optimality. Uniform weighting sets $w_{\text{base}} = 1$ to treat all valid supports equally.

- **Filtering (Hard vs. Soft)** Controls negative-gain handling. Hard filtering applies the mask $w_{\text{hard}} \leftarrow w_{\text{base}} \cdot \mathbb{I}_{\{\tilde{A} > 0\}}$ to strictly prune suboptimal transitions, while Soft filtering applies identity $w_{\text{soft}} \leftarrow w_{\text{base}}$ to retain full connectivity.

This formulation allows SPAR to adapt the stability-connectivity trade-off: selecting aggressive filtering for dense-reward tasks to maximize optimality, or conservative retention for sparse-reward tasks to maintain manifold connectivity.

### 3.2.3. RESIDUAL MODELING

We model the residual distribution via weighted reconstruction, where weights $w(s, a)$ emphasize high-value regions while preserving support geometry. The policy parameterization adapts to residual topology:

**Deterministic Regressor (SPAR-MLP).** For unimodal residuals, we employ a direct regressor $\pi_{\text{res}} : \mathcal{S} \times \mathcal{A} \to \mathbb{R}^d$ with total objective

$$\mathcal{L}_{\text{total}}^{\text{MLP}} = \mathbb{E}_{(s,a) \sim \mathcal{D}}\left[w(s, a) \cdot \|\pi_{\text{res}}(s, \pi_{\text{base}}(s)) - (a - \pi_{\text{base}}(s))\|_2^2\right] - \lambda_g \, \mathbb{E}_{s \sim \mathcal{D}}\left[Q_{\text{rob}}(s, a_{\text{synth}})\right], \quad (7)$$

where $a_{\text{synth}} = \pi_{\text{base}}(s) + \pi_{\text{res}}(s, \pi_{\text{base}}(s))$. The second term performs unconstrained gradient ascent on $Q_{\text{rob}}$ for policy improvement.

**Generative Model (SPAR-CVAE).** For multimodal residuals, we adopt a conditional VAE with encoder $q_\phi(z \mid s, \Delta a)$ and decoder $\pi_{\text{res}}(s, \pi_{\text{base}}(s), z)$, trained via weighted ELBO:

$$\mathcal{L}_{\text{fit}}^{\text{CVAE}} = \mathbb{E}_{(s,a) \sim \mathcal{D}}\left[w(s, a) \cdot \|\pi_{\text{res}}(s, \pi_{\text{base}}(s), z) - (a - \pi_{\text{base}}(s))\|_2^2\right] + \beta \cdot \text{KL}\left(q_\phi(z \mid s, \Delta a) \,\|\, p(z)\right), \quad (8)$$

where $z \sim q_\phi(\cdot \mid s, a - \pi_{\text{base}}(s))$. Policy improvement is achieved through latent self-imitation (Section 3.2.4), not direct gradient ascent. The advantage weight is applied only to the reconstruction term: it selects high-value residual actions to fit, while the KL term remains a global latent regularizer. Weighting the KL term would over-regularize high-advantage samples toward the prior and weaken the preservation of useful multimodal residual structure. We choose a CVAE for the local residual distribution around a frozen anchor; diffusion- or flow-based generators remain compatible alternatives, as examined in Appendix C.2.

### 3.2.4. LATENT SELF-IMITATION

We apply latent self-imitation to the residual policy of SPAR-CVAE, resulting in **SPAR-PROJ**. To resolve the fitting-improvement conflict, we propose latent self-imitation: a derivative-free method that guides policy improvement by a latent-sampling weighted regression mechanism. Due to its sampling requirement, this approach applies exclusively to generative policies and not to deterministic policies.

**Algorithm Design.** For state $s$, the CVAE decoder induces a residual manifold $\mathcal{M}_s = \{\pi_{\text{res}}(s, \pi_{\text{base}}(s), z) \mid z \in \mathcal{Z}\}$. We maintain a target policy $\pi'_{\text{res}}$ (EMA copy of $\pi_{\text{res}}$), sample $K$ latent candidates $\{z_k\}_{k=1}^K \sim p(z)$, and decode residuals $\Delta a_k = \pi'_{\text{res}}(s, \pi_{\text{base}}(s), z_k) \in \mathcal{M}_s$, which yield candidate actions $a_k = a_{\text{base}} + \Delta a_k$. Candidate advantages and normalized weights are computed using the frozen critic in Stage I (3.1):

$$\omega_k = \frac{w(\tilde{A}_k)}{\sum_j w(\tilde{A}_j)}, \quad \tilde{A}_k = \frac{Q_{\text{rob}}(s, a_k) - Q_{\text{rob}}(s, a_{\text{base}})}{\sigma_Q}. \tag{9}$$

The guidance loss regresses the student policy toward these weighted targets with stop-gradient applied to $\omega_k$ and $\Delta a_k$:

$$\mathcal{L}_{\text{guide}}(\theta) =$$
$$\mathbb{E}_{s \sim \mathcal{D}} \left[ \sum_{k=1}^K \omega_k \left\| \Delta a_{\text{res}}(s, \pi_{\text{base}}(s), z_k; \theta) - \Delta a_k \right\|_2^2 \right]. \tag{10}$$

**Geometric Safety.** The following proposition (the proof is in A.3) formalizes why latent self-imitation avoids off-manifold drift: by restricting updates to chords within the residual manifold, it achieves second-order drift suppression compared to the linear deviation of gradient ascent.

**Proposition 3.3.** *Assume the residual manifold $\mathcal{M}_s$ is $C^2$-smooth with maximum curvature $\kappa$. Let $d(x, \mathcal{M}_s)$ denote the Euclidean distance from point $x$ to $\mathcal{M}_s$.*

*(i)* **Convex Combination.** *For fixed samples $\{z_k, \Delta a_k, \omega_k\}$ with $\Delta a_k \in \mathcal{M}_s$ and $\sum_k \omega_k = 1$, the minimizer of loss (10) is the convex combination $\Delta a_{\text{res}}^*(s) = \sum_{k=1}^K \omega_k \Delta a_k$.*

*(ii)* **Second-Order Safety.** *By (i), updates for $K = 2$ lie on the chord connecting two manifold points. Specifically, the chord $x_\alpha = (1 - \alpha)x + \alpha y$ between $x, y \in \mathcal{M}_s$ satisfies*

$$\sup_{\alpha \in [0,1]} d(x_\alpha, \mathcal{M}_s) \leq \frac{\kappa}{8} \|y - x\|_2^2, \tag{11}$$

*By comparison, gradient ascent $x^+ = x + \eta v_{\text{grad}}$ with $v_{\text{grad}} \propto \nabla_a Q_{\text{rob}}$ generically yields linear drift $d(x^+, \mathcal{M}_s) = \Theta(\eta \|v_{\text{grad}}\|_2)$.*

According to this proposition, SPAR-PROJ resolves the fitting-improvement conflict by latent self-imitation.

### 3.3. Stage III: Deployment-Time Rectification

Finally, SPAR enforces baseline-anchored safety at inference. We accept the residual action $a_{\text{res}}$ only if the predicted baseline-relative improvement exceeds thresholds:

$$G(a_{\text{res}}) = \begin{cases} a_{res}, & \text{if } \Delta Q_{rob} > \eta_{\text{abs}} \wedge \dfrac{\Delta Q_{rob}}{|Q_{\text{rob}}^{\text{base}}| + \epsilon} > \eta_{\text{rel}} \\ 0, & \text{otherwise.} \end{cases} \tag{12}$$

We use the shared default thresholds $\eta_{\text{abs}} = 10^{-4}$ and $\eta_{\text{rel}} = 0.01$ across tasks, so Stage III acts as a conservative safety filter with fixed deployment settings.

## 4. Experiments

We evaluate SPAR on the D4RL (CORL version) benchmark (Fu et al., 2020). To rigorously isolate the contribution of the residual learning mechanism, we employ a fixed, standard Behavior Cloning (BC) policy as the anchor $\pi_\beta$ across all experiments. This design verifies whether SPAR can extract optimal performance solely through action rectification, independent of complex generative backbones. We focus on four research questions:

**Q1: Rectification Efficacy (Sec. 4.1).** Can SPAR consistently outperform strong offline baselines even when anchored to a simple, unimodal BC policy?

**Q2: Topology-Dependent Architecture (Sec. 4.2).** Does the residual distribution explain the advantages and disadvantages of SPAR's policy modeling?

**Q3: Support-Preserving Improvement (Sec. 4.3).** Does the latent self-imitation effectively resolve the gradient conflict between reward maximization and support constraints?

**Q4: Component Sensitivity (Sec. 4.4).** What is the impact of adjustable conservatism, sampling range, and sample weighting mode on performance?

We evaluate SPAR across three distinct D4RL domains:

- MuJoCo Locomotion: The medium-replay (MR) and medium-expert (ME) datasets for HalfCheetah (HC), Hopper (HP), and Walker2d (WK).
- AntMaze(AM) Navigation: The sparse-reward Umaze-diverse(UD) and Large-diverse(LD) environments.
- Adroit Manipulation: The high-dimensional Pen-Cloned (Pen-CL) and Pen-Human (Pen-HM) tasks.

### 4.1. Comparative Evaluation on D4RL

**Baselines and Implementation.** We report normalized scores from the D4RL benchmark. Table 1 lists both vari-

ants across all tasks: SPAR-MLP is a compact residual model that works well on simple unimodal residuals, and SPAR-PROJ is the robust default for multimodal, sparse, or uncertain residual geometry. Hyperparameters are listed in Appendix C.1.

We categorize baselines into two paradigms based on their dependence on value function gradients for policy extraction: Gradient-Based Optimization: BCQ(Fujimoto et al., 2019) TD3+BC(Fujimoto & Gu, 2021), CQL(Kumar et al., 2020), PLAS(Zhou et al., 2021), Diff-QL(Wang et al., 2022). In-Sample Learning: AWAC(Nair et al., 2020), IQL(Kostrikov et al., 2021), IDQL(Hansen-Estruch et al., 2023), LAPO(Chen et al., 2022), EQL(Xu et al., 2023), CQL-AW(Hong et al., 2023). Table 1 presents the evaluation results. We analyze performance trends across three distinct geometric regimes. Additional evaluations on recent baselines, Stage III threshold sensitivity, inference cost, flow-based variants, and MuJoCo medium results are reported in Appendix C.2.

**MuJoCo Locomotion.** On medium-replay datasets, SPAR-MLP demonstrates a significant advantage. It consistently outperforms In-Support methods (e.g., IQL, AWAC) and Policy Gradient Guidance with only a single policy network. Notably, it aligns with or even exceeds the performance of generative models such as Diffusion-QL and IDQL, while avoiding significant computational overhead. Conversely, the performance of SPAR-PROJ on these simple tasks is limited by the stochasticity of its outputs and the constraints imposed by conservative improvement. However, in medium-expert regimes, the deterministic SPAR-MLP exhibits a marked performance drop. In contrast, SPAR-PROJ successfully recovers optimal performance in these multimodal tasks.

**Adroit Manipulation.** In high-dimensional manipulation tasks such as Adroit Pen, SPAR-PROJ demonstrates superior stability compared to Advantage-Weighted and Constraint-based baselines. Achieving leading scores on both Pen-Cloned and Pen-Human tasks, SPAR-PROJ proves its capability to handle complex control problems in high-dimensional state spaces without collapsing on the narrow feasible manifolds, a common failure mode for traditional methods.

**AntMaze Navigation.** In sparse-reward navigation tasks, SPAR-PROJ establishes a robust performance profile. It consistently surpasses policy gradient guidance and In-Support methods while maintaining strong competitiveness against computation-heavy Generative baselines. Especially in long-horizon tasks like AntMaze-Large, SPAR-PROJ demonstrated exceptional in-neighborhood recovery capabilities, thereby enhancing the robustness of policy stitching.

### 4.2. Mechanism Analysis Based on Residual Demonstration

We report both SPAR-MLP and SPAR-PROJ across all 10 environments in Table 1 and visualize representative residual distributions reduced by the BC baseline in Figure 2. These results reveal the intrinsic connection between residual geometry and policy modeling. **Residual Geometry and Policy Model Choice.** As illustrated in Figure 2, the residual distribution exhibits unimodal characteristics in MuJoCo medium-replay environments, while displaying multimodal mixtures in remaining datasets.

In unimodal regimes, SPAR-MLP demonstrates significant modeling superiority. Compared to In-Support methods, which suffer from excessive conservatism due to weighted regression, SPAR-MLP maintains appropriate extrapolation capabilities. Compared to Policy Guidance methods, the backbone network of SPAR-MLP focuses on fitting, allowing for aggressive extrapolation within the local residual region. By leveraging the smooth extrapolation property of MuJoCo MR, this approach avoids over-conservatism and prevents model collapse caused by conflicting gradients acting on the backbone. Crucially, compared to SPAR-PROJ, SPAR-MLP avoids the unnecessary variance and optimization complexity introduced by stochastic sampling, as unimodal optimization is fundamentally a function approximation problem rather than a density estimation one.

In multimodal datasets, the core failure mode of SPAR-MLP stems from its MSE objective approximating the conditional expectation $\mathbb{E}[\Delta a|s]$, causing the mean action to fall into low-density or invalid regions. While Generative Models provide multimodal initialization, their unconstrained gradients still risk out-of-distribution (OOD) drift. SPAR-PROJ addresses this by modeling $p(\Delta a|s)$ via a CVAE combined with latent self-imitation, effectively capturing multimodal distributions while geometrically precluding OOD risks.

**Difficult Extrapolation in Narrow Feasible Regions.** In high-dimensional tasks such as Adroit Pen, the expert data distribution is extremely narrow and sparse, covering only a thin slice of the valid state-action space. Standard baselines often fail in this context because their value gradients blindly extrapolate into out-of-distribution (OOD) regions—areas that appear to offer high returns but are dynamically unstable, thus necessitating a highly cautious improvement mechanism.

**Branching Ambiguity in AntMaze.** In the first stage, we increased the expectile to restore the critic's discriminative capability in these extremely sparse-reward environments. We then observed that the distribution of recovered high-value trajectories in AntMaze exhibits a structural sparsity similar to Adroit, but stemming from the discrete branching points inherent to navigation. This pronounced multimodal-

*Table 1.* Normalized scores on D4RL benchmarks. We report the mean and standard deviation over 3 seeds. **Bold** indicates the highest mean score. MuJoCo MR means MuJoCo Medium-Replay, MuJoCo ME means MuJoCo Medium-Expert.

| Domain | Task | Policy Gradient Guidance | | | | | In-Support Learning | | | | | | Ours | | |
|---|---|---|---|---|---|---|---|---|---|---|---|---|---|---|---|
| | | BCQ | TD3+BC | CQL | PLAS | Diff-QL | AWAC | IQL | LAPO | IDQL | EQL | CQL-AW | Base | SPAR-MLP | SPAR-PROJ |
| MuJoCo MR | HalfCheetah | 40.4 | 44.6 | 45.5 | 45.7 | 47.8 | 40.5 | 44.2 | 41.9 | 45.1 | 47.2 | 49.0 | 40.5 | **50.9** $\pm$ 0.6 | 43.6 $\pm$ 0.2 |
| | Hopper | 53.3 | 60.9 | 95.0 | 51.9 | 101.3 | 37.2 | 94.7 | 50.1 | 82.4 | 74.6 | 71.0 | 42.9 | **101.9** $\pm$ 0.4 | 70.4 $\pm$ 8.9 |
| | Walker2d | 52.1 | 81.8 | 81.6 | 14.3 | **95.5** | 27.0 | 73.9 | 60.6 | 79.8 | 83.2 | 83.0 | 53.3 | 94.0 $\pm$ 0.7 | 72.4 $\pm$ 2.6 |
| MuJoCo ME | HalfCheetah | 89.1 | 90.7 | 91.6 | 94.3 | 96.8 | 42.8 | 86.7 | 94.2 | 94.4 | 90.6 | 84.0 | 63.1 | 85.2 $\pm$ 10.7 | **97.0** $\pm$ 0.1 |
| | Hopper | 81.8 | 98.0 | 105.4 | 94.7 | **111.1** | 55.8 | 91.5 | 111.0 | 105.3 | 105.5 | 91.0 | 55.2 | 1.1 $\pm$ 0.2 | 108.7 $\pm$ 1.9 |
| | Walker2d | 109.0 | 110.1 | 108.8 | 97.2 | 110.9 | 74.5 | 109.6 | 110.9 | 111.6 | 110.2 | 109.0 | 109.7 | 0.0 $\pm$ 0.3 | **113.4** $\pm$ 0.8 |
| Adroit | Pen-Cloned | 55.0 | 71.4 | 40.3 | 49.0 | 66.2 | 49.3 | 62.2 | 55.0 | 63.6 | 46.9 | -2.5 | 50.4 | 0.1 $\pm$ 0.2 | **76.2** $\pm$ 3.5 |
| | Pen-Human | 2.2 | 0.0 | 14.9 | 49.8 | 56.6 | 1.0 | 47.5 | 2.2 | 49.8 | 44.3 | -3.0 | 53.5 | 0.1 $\pm$ 0.4 | **62.7** $\pm$ 2.2 |
| AntMaze | Umaze-Div | 28.0 | 1.7 | 39.2 | 62.0 | 57.3 | -0.8 | 37.3 | 28.0 | 62.0 | 50.8 | 54.0 | 40 | 63.3 $\pm$ 7.6 | **76.7** $\pm$ 10.4 |
| | Large-Div | 12.3 | -3.7 | 37.5 | 45.3 | **72.8** | 4.3 | 45.5 | 12.3 | 56.4 | 38.0 | 40.0 | 20 | 15.0 $\pm$ 5.0 | 71.0 $\pm$ 7.4 |

*Table 2.* Component ablation of SPAR.

| Task | CVAE base | Global SI | Res. w/o LSI | w/o Stage III | SPAR full |
|---|---|---|---|---|---|
| HP-MR | 28.2 | 12.1 | 72.7 | 49.5 | **101.9** |
| HC-ME | 60.3 | 1.6 | 94.1 | 96.1 | **97.0** |
| Pen-CL | 58.8 | 3.9 | 67.8 | 61.6 | **76.2** |
| AM-LD | 15.0 | 0.0 | 40.0 | 60.0 | **71.0** |

ity renders traditional non-generative methods susceptible to mode-averaging, often resulting in high variance and ineffective exploration. Unlike computationally expensive generative models, SPAR-PROJ operates as an efficient local mode-selector. By resolving directional ambiguity at these branching points, it enforces convergence to a single valid mode, thereby enabling precise trajectory stitching without the high inference overhead of diffusion models.

**Component Necessity and Support Preservation.** The residual-geometry analysis above explains why the residual policy architecture should adapt to the topology of $\Delta a = a - \pi_{\text{base}}(s)$. This motivates two empirical checks: whether the full rectification pipeline is necessary, and whether generated actions stay close to the empirical action support. Table 2 directly ablates the three key components of SPAR. Global-space self-imitation collapses on the harder tasks, confirming that improvement must be constrained to the anchored residual space. Residual learning without latent self-imitation and removing Stage III rectification both provide partial gains; the full pipeline performs best overall. Figure 3 gives a direct support check on Pen-Cloned. The boundary plot uses a shared PCA projection for visualization. The q95 kNN support-distance ratio is computed in the original action space and normalized by the dataset boundary: SPAR-PROJ scores 0.98, and SPAR-PLAS scores 6.32.

### 4.3. Mechanism Analysis: Guidance without Drift

A central theoretical assertion of our framework is that the update direction maximizing the Q-value often pushes the policy towards regions outside the data support. To verify this phenomenon, we conducted an empirical analysis focusing on two key dimensions:

**Quantifying Gradient Conflict.** To measure the tension

between reward maximization and distribution fitting, we introduced two diagnostic metrics based on the real training configuration (Adam optimizer, learning rate $3 \times 10^{-4}$). Let $\mathcal{L}_{\text{fit}}$ denote the distribution fitting loss, $\Phi_\theta$ denote a generic network module subject to value gradients and $\Delta\theta$ be the actual parameter update vector induced by a single step of value guidance.

We define $\text{DD} = \langle \nabla_\theta \mathcal{L}_{\text{fit}}, \Delta\theta \rangle$ as the directional projection of the update vector onto the fitting gradient, where a positive value indicates an update direction that opposes the fitting objective (i.e., damage to the fit). Complementarily, we let $\text{VSD} = \mathcal{L}_{\text{fit}}(\theta + \Delta\theta) - \mathcal{L}_{\text{fit}}(\theta)$ quantify the actual degradation in fitting performance after a single update step. Note that $\Phi_\theta$ does not necessarily represent the direct action-generation policy; for instance, it could be the latent actor in PLAS(Zhou et al., 2021) or the noise prediction network in Diffusion-QL. However, via the chain rule, gradients from both the fitting and improvement objectives are backpropagated to this common set of parameters $\theta$. Consequently, these gradients can be directly compared within this parameter space, allowing us to identify and analyze any conflicts in their update directions.

As shown in Table 3, baseline methods (SPAR-MLP, SPAR-PLAS) exhibit large positive values on both DD and VSD, and SPAR-PROJ reduces both metrics by multiple orders of magnitude. This indicates that, under standard learning rates, gradient-based guidance induces severe parameter drift and rapidly compromises the policy's distributional consistency. SPAR-PROJ's Latent Self-Imitation filters out the distribution-destroying components of the gradient, preserves the value signal, and nearly eliminates deviation caused by optimizer step sizes, empirically validating Proposition 3.3.

**Ablation Study on Guidance Weight.** Table 4 ablates the weight of the value guidance objective. We implement SPAR-PLAS based on PLAS(Zhou et al., 2021), which is a generative residual policy model guided by gradient updates.

For gradient-based methods (SPAR-MLP, SPAR-PLAS), while increasing the guidance weight may initially improve

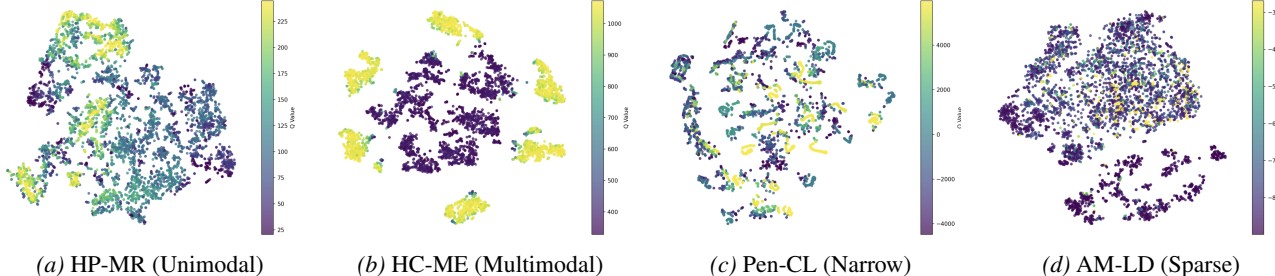

*(a)* HP-MR (Unimodal)  *(b)* HC-ME (Multimodal)  *(c)* Pen-CL (Narrow)  *(d)* AM-LD (Sparse)

*Figure 2.* Visualization of the $(a|s)$ distribution for Residuals, generated by projecting 4000 sampled actions into a 2D space via joint t-SNE, with points colored by Q-values.

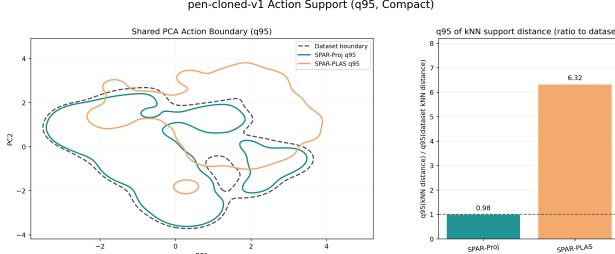

*Figure 3.* Action-support diagnostic on Pen-Cloned. Left: q95 action boundary in a shared PCA space for visualization. Right: q95 kNN support-distance ratio computed in the original action space and normalized by the dataset boundary. SPAR-PROJ stays close to empirical support; SPAR-PLAS deviates substantially.

*Table 3.* Gradient Conflict Metrics across D4RL benchmarks. SPAR-PROJ significantly reduces conflict metrics compared to baseline architectures.

| Task | Metric | SPAR-MLP | SPAR-PLAS | SPAR-PROJ |
|------|--------|----------|-----------|-----------|
| HP-MR | DD | 2.4e-3 | 9.2e-3 | **1.0e-10** |
|       | VSD | 3.2e-3 | 1.8e-4 | **7.2e-5** |
| HC-ME | DD | 2.8e-3 | 1.2e-2 | **3.6e-6** |
|       | VSD | 8.0e-4 | 2.9e-4 | **3.2e-6** |
| Pen-CL | DD | 3.6e-4 | 5.8e-3 | **2.2e-5** |
|        | VSD | 4.5e-3 | 6.4e-5 | **1.6e-5** |
| AM-LD | DD | 1.6e-3 | 7.9e-3 | **1.9e-5** |
|       | VSD | 6.0e-3 | 4.0e-3 | **1.5e-4** |

scores, excessive gradient steps forcibly push the policy out of the data support, ultimately degrading performance.

Since SPAR-PROJ acquires high-value samples by sampling within the support, its value improvement is constrained to the data distribution. Consequently, the value guidance objective safely enhances performance and demonstrates strong robustness to variations in guidance weight.

Table 4 also shows that, in environments with dense data coverage, all methods are largely insensitive to variations in the guidance weight $\lambda_g$. In settings with narrow data distributions, SPAR-PROJ maintains the strongest robustness. This supports conservative policy optimization under sparse and narrow effective support, where aggressive exploitation can trigger catastrophic OOD drift, and it further confirms

*Table 4.* Ablation study on guide weight $\lambda_g$. We compare SPAR-PROJ with methods with gradient conflicts (SPAR-MLP, SPAR-PLAS) across different tasks.

| Task | Method | $\lambda_g = 0$ | $\lambda_g = 0.5$ | $\lambda_g = 2$ |
|------|--------|------|--------|------|
| HP-MR | SPAR-MLP | 72.71 | **101.9** | 101.5 |
|       | SPAR-PROJ | 67.1 | 70.4 | 54.3 |
| HC-ME | SPAR-PLAS | 91.3 | 92.4 | 92.6 |
|       | SPAR-PROJ | 94.05 | **97.0** | 96.6 |
| Pen-CL | SPAR-PLAS | 72.1 | 74.1 | 64.2 |
|        | SPAR-PROJ | 67.8 | **76.2** | 65.3 |
| AM-LD | SPAR-PLAS | 33.3 | 21.6 | 35.0 |
|       | SPAR-PROJ | 40.0 | **71.0** | 57.5 |

the stability of Latent Self-Imitation.

### 4.4. Analysis of Weighting Schemes and Uncertainty

**Ablation Study on Weighting Schemes.** Table 5 reveals a task-dependent trade-off between Soft and Hard filtering strategies. In dense-coverage MuJoCo tasks, both strategies perform comparably, with Hard achieving marginally higher scores on HC-ME due to abundant data ensuring connectivity after pruning. In sparse-reward AntMaze-Large, Soft demonstrates decisive superiority because trajectory stitching requires preserving low-advantage transitions that bridge topologically disconnected segments. Hard filtering can sever these bridges when advantage estimates fluctuate near zero. In narrow-distribution Pen-Cloned, Soft achieves the global peak by preserving low-advantage transitions that enable gradual exploration toward sparse high-value regions. Hard filtering discards these transitions via binary thresholding, limiting peak exploitation despite more stable performance across temperatures. The temperature $T$ controls the sharpness of Boltzmann weighting. At $T = 0.3$, Soft strikes an optimal balance between connectivity preservation and value focus; Hard lacks this adaptive granularity. Advantage-Weighted Inference consistently outperforms uniform sampling across all tasks, confirming that Soft Weighting robustly extracts high-value actions while respecting the geometric constraints of the residual support.

**Ablation Study on Uncertainty Weight.** Table 6 analyzes the impact of the uncertainty penalty $\lambda_u$. On dense-coverage

*Table 5.* Ablation study on unified adaptive weighting: $T$ in $w_{\text{base}} = \exp(\tilde{A}/T)$, and soft/hard filtering choice.

| Filter | $T$ | HP-MR | HC-ME | Pen-CL | AM-LD |
|---|---|---|---|---|---|
| Soft | 0.1 | 88.7 | 90.9 | 56.9 | 48.3 |
| | 0.3 | 94.9 | 88.4 | **76.2** | **71.0** |
| | 1.0 | 101.5 | 88.6 | 56.3 | 65.0 |
| | Uni | 99.0 | 87.8 | 56.4 | 45.0 |
| Hard | 0.1 | 90.9 | 96.0 | 72.1 | 17.5 |
| | 0.3 | 97.1 | **97.0** | 65.4 | 30.0 |
| | 1.0 | **101.9** | 96.1 | 65.5 | 40 |
| | Uni | 98.6 | 93.3 | 64.8 | 32.5 |

*Table 6.* Ablation study on uncertainty weight $\lambda_u$.

| Task | Method | $\lambda_u = 0$ | $\lambda_u = 0.5$ | $\lambda_u = 2$ |
|---|---|---|---|---|
| HP-MR | SPAR-MLP | 55.2 | **101.9** | 93.2 |
| HC-ME | SPAR-PROJ | 91.3 | **97.0** | 96.5 |
| Pen-CL | SPAR-PROJ | 39.2 | **76.2** | 73.0 |
| AM-LD | SPAR-PROJ | 55.0 | **71.0** | 60.0 |

tasks like HP-MR and HC-ME, moderate $\lambda_u$ yields optimal performance, while removing the penalty causes noticeable degradation, particularly on HP-MR, where over-optimistic value extrapolation leads to unsafe actions despite abundant data. On support-limited tasks like Pen-Cloned and AntMaze-Large-Diverse, the penalty becomes critical: performance collapses from 76.2 to 39.2 and from 71.0 to 55.0 respectively when $\lambda_u = 0$, as the policy aggressively exploits poorly estimated regions outside the narrow residual manifold. Conversely, excessive pessimism ($\lambda_u = 2$) also harms performance on AntMaze, suppressing legitimate exploration across sparse waypoints. These results show that uncertainty-aware weighting adaptively modulates the trade-off between exploitation and conservatism.

## 5. Conclusion

**Conclusion.** We introduced Support-Preserving Action Rectification (SPAR), a framework that reframes offline policy improvement as localized residual correction anchored to a frozen behavior cloning policy. This design contracts the effective search space while preserving data manifold constraints. To resolve the fitting-optimization conflict in residual space, SPAR employs Latent Self-Imitation, a derivative-free mechanism that replaces unstable value-gradient ascent with latent-sampling weighted regression. The analysis explains how residual search-space contraction and latent self-imitation suppress off-manifold drift, and D4RL evaluations show strong performance across locomotion, manipulation, and sparse-reward navigation tasks.

**Limitations and Future Work.** SPAR's effectiveness still depends on the quality of the frozen anchor and the reliability of conservative value estimates, especially in sparse-reward regimes. The generative residual variant also incurs higher inference cost than deterministic residual regression,

although the appendix measurements show lower latency and memory than iterative diffusion-based alternatives. Future work includes broader benchmark validation, automatic diagnostics for residual topology, and stronger offline RL guarantees beyond the mechanism-level analysis developed here.

## Acknowledgements

This work was supported in part by the Key R&D Program of Zhejiang Province (2025C01212), in part by the National Natural Science Foundation of China (Grant No. 62273303), and in part by the Yongjiang Talent Introduction Programme (2022A-240-G).

## Impact Statement

This paper presents work whose goal is to advance the field of Machine Learning. There are many potential societal consequences of our work, none which we feel must be specifically highlighted here.

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

# A. Proofs

## A.1. Proof of Theorem 3.1

We first restate the theorem for completeness.

**Theorem A.1** (Theorem 3.1, restated). *Under L-Lipschitz continuity of $Q(s, \cdot)$, $\sigma$-sub-Gaussian value observations, and action diameter $D$, the effective data requirement for $\epsilon$-optimal action identification within the $\delta_\rho$-neighborhood with probability at least $1 - \beta$ satisfies*

$$N(\epsilon, \Omega) = \tilde{\Theta}\left(\frac{\sigma^2}{\epsilon^2} \cdot \mathcal{N}(\Omega, \tfrac{\epsilon}{2L}) \cdot \frac{d}{\beta} \cdot \log D\right), \tag{13}$$

*where $\tilde{\Theta}$ absorbs logarithmic factors including $\log \mathcal{N}(\Omega, \epsilon/(2L))$.*

This analysis explains the statistical effect of residual search-space contraction; it is not intended as a complete offline RL performance guarantee under arbitrary distribution shift.

*Proof.* Fix a state $s$ and write $f(a) \triangleq Q(s, a)$ for brevity. Let $\Omega \subseteq \mathbb{R}^d$ be compact. By construction of $\delta_\rho$ as the $(1 - \rho)$-quantile with small $\rho$ (e.g., $\rho = 0.05$), the residual region $\Omega_{\mathrm{res}}$ contains $1 - \rho$ fraction of the dataset mass. This ensures that each $\epsilon/(2L)$-ball intersecting $\Omega_{\mathrm{res}}$ has sufficient samples for concentration when $\rho$ is small (Figure 4).

**Step 1: Covering at resolution $r = \epsilon/(2L)$.** Let $\mathcal{C} = \{c_1, \ldots, c_N\}$ be an $r$-cover of $\Omega$ with $n = \mathcal{N}(\Omega, r)$. By Lipschitz continuity, for any $a \in \Omega$ there exists $c_i \in \mathcal{C}$ such that

$$|f(a) - f(c_i)| \leq L\|a - c_i\|_2 \leq Lr = \epsilon/2. \tag{14}$$

**Step 2: Uniform concentration over the cover.** For each cover point $c_i$, form the empirical mean $\widehat{f}(c_i)$ from $m$ samples. Since observations are $\sigma$-sub-Gaussian, Hoeffding's inequality gives

$$\mathbb{P}\big(|\widehat{f}(c_i) - f(c_i)| \geq \epsilon/4\big) \leq 2\exp\big(-m\epsilon^2/(32\sigma^2)\big) = \frac{\beta}{n}. \tag{15}$$

Setting $m = (32\sigma^2/\epsilon^2)\log(2n/\beta)$ and applying a union bound yields

$$\mathbb{P}\Big(\max_{i \in [n]} |\widehat{f}(c_i) - f(c_i)| \leq \epsilon/4\Big) \geq 1 - \beta. \tag{16}$$

**Step 3: Constructing an $\epsilon$-optimal action.** Let $a^\star \in \arg\max_{a \in \Omega} f(a)$ and $c^\star \in \mathcal{C}$ its nearest cover point. On the event (16),

$$
\begin{aligned}
f(a^\star) &\leq f(c^\star) + \epsilon/2 && \text{(by (14))} && \text{(17)}\\
&\leq \widehat{f}(c^\star) + 3\epsilon/4 && \text{(by (16))} && \text{(18)}\\
&\leq \widehat{f}(\widehat{c}) + 3\epsilon/4 && \text{(where } \widehat{c} = \arg\max_{c_i} \widehat{f}(c_i)) && \text{(19)}\\
&\leq f(\widehat{c}) + \epsilon && \text{(by (16)).} && \text{(20)}
\end{aligned}
$$

Thus $\widehat{c}$ is $\epsilon$-optimal with probability $\geq 1 - \beta$.

**Step 4: Total data requirement.** The total number of samples required is $n \cdot m = \mathcal{N}(\Omega, \epsilon/(2L)) \cdot (32\sigma^2/\epsilon^2)\log(2\mathcal{N}(\Omega, \epsilon/(2L))/\beta)$. Absorbing dimension-independent logarithmic factors into $\tilde{\Theta}$ yields the stated scaling.

**Step 5: Geometric efficiency gain.** For convex regions (e.g., $\mathcal{A} = [-1, 1]^d$), the diameter bound $\mathcal{N}(\Omega, r) \leq (1 + D/r)^d$ holds (Vershynin, 2018). Applying this to $\Omega_{\mathrm{global}} = \mathcal{A}$ and $\Omega_{\mathrm{res}}$ with $r = \epsilon/(2L)$ gives

$$\mathcal{N}(\Omega_{\mathrm{global}}, r) \leq \left(1 + \frac{2LD_{\mathcal{A}}}{\epsilon}\right)^d, \tag{21}$$

$$\mathcal{N}(\Omega_{\mathrm{res}}, r) \leq \left(1 + \frac{4L\delta_\rho}{\epsilon}\right)^d. \tag{22}$$

The efficiency ratio satisfies

$$\frac{N_{\text{global}}}{N_{\text{spar}}} = \Theta\left(\left(\frac{D_{\mathcal{A}}}{2\delta_\rho}\right)^d \cdot \frac{\log(1 + 2LD_{\mathcal{A}}/\epsilon)}{\log(1 + 4L\delta_\rho/\epsilon)}\right). \tag{23}$$

In the regime $\epsilon < L\delta_\rho < LD_{\mathcal{A}}$, the logarithmic factor is bounded by a small constant ($< 2$ for D4RL tasks), so the dominant term is $(D_{\mathcal{A}}/(2\delta_\rho))^d$. $\qquad\square$

## A.2. Proof of Lemma 3.2

**Lemma A.2** (Lemma 3.2, restated). *Let $a^\mu(s) \in \arg\max_{a \in \text{supp}(\mu(\cdot|s))} Q(s, a)$. Under L-Lipschitz continuity of $Q(s, \cdot)$,*

$$\varepsilon_{\text{loc}}(s; \delta_\rho) \leq L \cdot \left[\|a^\mu(s) - \pi_{\text{base}}(s)\|_2 - \delta_\rho\right]_+. \tag{24}$$

*Proof.* Fix state $s$ and denote $a_{\text{base}} = \pi_{\text{base}}(s)$, $a^\mu = a^\mu(s)$. Consider two cases:

**Case 1:** $\|a^\mu - a_{\text{base}}\|_2 \leq \delta_\rho$. Then $a^\mu$ lies within the localized region $\{a : \|a - a_{\text{base}}\|_2 \leq \delta_\rho\}$, so

$$\max_{\|a - a_{\text{base}}\|_2 \leq \delta_\rho} Q(s, a) \geq Q(s, a^\mu). \tag{25}$$

By definition (5), $\varepsilon_{\text{loc}}(s; \delta_\rho) \leq 0$. Since $\varepsilon_{\text{loc}} \geq 0$ by construction, we have $\varepsilon_{\text{loc}} = 0$, matching the bound as $[\cdot]_+ = 0$.

**Case 2:** $\|a^\mu - a_{\text{base}}\|_2 > \delta_\rho$. Define the radial projection onto the $\delta_\rho$-ball:

$$\widetilde{a} = a_{\text{base}} + \delta_\rho \cdot \frac{a^\mu - a_{\text{base}}}{\|a^\mu - a_{\text{base}}\|_2}. \tag{26}$$

Then $\|\widetilde{a} - a_{\text{base}}\|_2 = \delta_\rho$, so $\widetilde{a}$ is feasible for the localized maximization. Therefore,

$$\max_{\|a - a_{\text{base}}\|_2 \leq \delta_\rho} Q(s, a) \geq Q(s, \widetilde{a}). \tag{27}$$

Using the definition (5) and $L$-Lipschitz continuity,

$$\varepsilon_{\text{loc}}(s; \delta_\rho) = Q(s, a^\mu) - \max_{\|a - a_{\text{base}}\|_2 \leq \delta_\rho} Q(s, a) \tag{28}$$

$$\leq Q(s, a^\mu) - Q(s, \widetilde{a}) \tag{29}$$

$$\leq L\|a^\mu - \widetilde{a}\|_2 \tag{30}$$

$$= L\big(\|a^\mu - a_{\text{base}}\|_2 - \delta_\rho\big), \tag{31}$$

where the last equality holds because $\widetilde{a}$ lies on the line segment between $a_{\text{base}}$ and $a^\mu$.

Combining both cases yields the stated bound. $\qquad\square$

## A.3. Proof of Proposition 3.3

We first restate the proposition for completeness.

**Proposition A.3** (Proposition 3.3, restated). *Assume the residual manifold $\mathcal{M}_s$ is $C^2$-smooth with maximum curvature $\kappa$. Let $d(x, \mathcal{M}_s) = \inf_{y \in \mathcal{M}_s} \|x - y\|_2$ denote the Euclidean distance to $\mathcal{M}_s$.*

*(i) **Convex Combination.** For fixed samples $\{z_k, \Delta a_k, \omega_k\}$ with $\Delta a_k \in \mathcal{M}_s$ and $\sum_k \omega_k = 1$, the minimizer of loss (10) is the convex combination $\Delta a_{\text{res}}^*(s) = \sum_{k=1}^K \omega_k \Delta a_k$.*

*(ii) **Second-Order Safety.** When $K = 2$, the chord $x_\alpha = (1 - \alpha)x + \alpha y$ between $x, y \in \mathcal{M}_s$ satisfies*

$$\sup_{\alpha \in [0,1]} d(x_\alpha, \mathcal{M}_s) \leq \frac{\kappa}{8}\|y - x\|_2^2, \tag{32}$$

*By comparison, gradient ascent $x^+ = x + \eta v_{\text{grad}}$ with $v_{\text{grad}} \propto \nabla_a Q_{\text{rob}}$ generically yields linear drift $d(x^+, \mathcal{M}_s) = \Theta(\eta\|v_{\text{grad}}\|_2)$.*

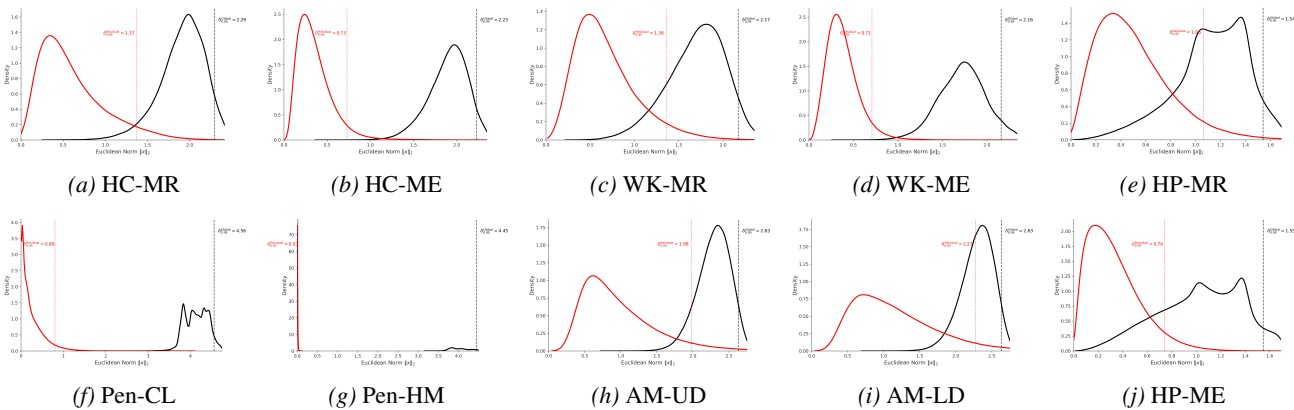

*Figure 4.* Visualization of the size of the action space. The red curve represents the distribution of the residual action space size, while the black curve represents the distribution of the global action space size of the dataset. To mitigate the impact of long-tail distributions, we adopt the 95th percentile of $D_a$ as the effective distance threshold.

*Proof.* **Part (i): Convex combination minimizer.** Fix samples $\{z_k, \Delta a_k, \omega_k\}$ and consider the pointwise quadratic objective

$$\mathcal{L}(\Delta a) = \sum_{k=1}^{K} \omega_k \|\Delta a - \Delta a_k\|_2^2. \tag{33}$$

Taking the gradient with respect to $\Delta a$ and setting to zero yields

$$\nabla_{\Delta a}\mathcal{L} = 2\sum_{k=1}^{K} \omega_k(\Delta a - \Delta a_k) = 0 \implies \Delta a^* = \sum_{k=1}^{K} \omega_k \Delta a_k, \tag{34}$$

where we used $\sum_k \omega_k = 1$. Since each $\Delta a_k \in \mathcal{M}_s$ and $\omega_k \geq 0$, the solution is a convex combination of valid manifold points.

**Part (ii): Second-order chord deviation.** Let $x, y \in \mathcal{M}_s$ and define the chord $x_\alpha = (1-\alpha)x + \alpha y$ for $\alpha \in [0, 1]$. By the $C^2$-smoothness of $\mathcal{M}_s$ and the uniform curvature bound $\kappa$, the second fundamental form satisfies $\|II_u(v, v)\|_2 \leq \kappa\|v\|_2^2$ for all tangent vectors $v$ (do Carmo, 1992). Consequently, the normal component of the chord vector is bounded by (Absil & Malick, 2012)

$$\|(I - P_x)(y - x)\|_2 \leq \frac{\kappa}{2}\|y - x\|_2^2, \tag{35}$$

where $P_x$ projects onto the tangent space $T_x\mathcal{M}_s$. Applying Taylor expansion along the chord and using the curvature bound yields the uniform deviation bound

$$\sup_{\alpha \in [0,1]} d(x_\alpha, \mathcal{M}_s) \leq \frac{\kappa}{8}\|y - x\|_2^2. \tag{36}$$

This establishes that chord-based updates incur only *quadratic* off-manifold drift controlled by curvature.

**Gradient ascent drift.** For a gradient update $x^+ = x + \eta v_{\text{grad}}$ with $v_{\text{grad}} \propto \nabla_a Q_{\text{rob}}$, decompose $v_{\text{grad}} = v_{\text{grad}}^\top + v_{\text{grad}}^\perp$ into tangent and normal components relative to $T_x\mathcal{M}_s$. Since the value gradient is generically not tangent to the manifold (i.e., $\|v_{\text{grad}}^\perp\|_2 > 0$), the distance after update satisfies

$$d(x^+, \mathcal{M}_s) = \eta\|v_{\text{grad}}^\perp\|_2 + o(\eta) = \Theta(\eta\|v_{\text{grad}}\|_2), \tag{37}$$

which is *linear* in the stepsize $\eta$. This contrasts sharply with the quadratic chord deviation.

**Geometric implication.** When the chord length $\|y - x\|_2$ is comparable to a small gradient step ($\|y - x\|_2 = O(\eta)$), latent self-imitation confines updates to an $O(\eta^2)$-tube around $\mathcal{M}_s$. Gradient ascent induces $O(\eta)$ drift. This quadratic suppression of off-manifold deviation underpins the support-preserving property of SPAR-PROJ. $\square$

## B. Algorithm Pseudocode

We present the complete SPAR pipeline in three stages. Stage I trains a conservative anchor policy and critic ensemble. Stage II learns a residual policy with topology-adaptive architectures: SPAR-MLP for unimodal residuals, SPAR-PLAS for gradient-guided latent policies, and SPAR-PROJ for projection-based latent self-imitation. Stage III applies value-gated rectification at inference.

---

**Algorithm 1** Stage I

---

**Require:** Dataset $\mathcal{D}$, expectile $\tau = 0.5$, uncertainty weight $\lambda_u$
**Ensure:** Base policy $\pi_{\text{base}}$, robust critic $Q_{\text{rob}}$
1: Initialize BC policy $\pi_{\text{base}}$ and critic ensemble $\{Q_{\theta_i}\}_{i=1}^{M}$
2: **for** each gradient step **do**
3:     Sample batch $\{(s, a, r, s')\} \sim \mathcal{D}$
4:     $\mathcal{L}_{\text{BC}} \leftarrow \mathbb{E}[\|\pi_{\text{base}}(s) - a\|_2^2]$ {Pure BC}
5:     Update $\pi_{\text{base}}$ via $\nabla_\phi \mathcal{L}_{\text{BC}}$
6:     Compute target: $y \leftarrow r + \gamma \cdot \frac{1}{M} \sum_i Q_{\theta_i}(s', \pi_{\text{base}}(s'))$
7:     $\mathcal{L}_Q \leftarrow \mathbb{E}[\rho_\tau(y - Q_{\theta_i}(s, a))]$ {Expectile loss}
8:     Update critics via $\nabla_{\theta_i} \mathcal{L}_Q$
9: **end for**
10: Freeze $\pi_{\text{base}}$; compute $Q_{\text{rob}}(s, a) \leftarrow \mu_Q(s, a) - \lambda_u \sigma_Q(s, a)$
11: **return** $\pi_{\text{base}}, Q_{\text{rob}}$

---

**Algorithm 2** SPAR-MLP

---

**Require:** $\pi_{\text{base}}, Q_{\text{rob}}, \mathcal{D}$, temperature $T$, guide weight $\lambda_g$
**Ensure:** Deterministic residual $\pi_{\text{res}}$
1: Initialize MLP $\pi_{\text{res}} : \mathcal{S} \times \mathcal{A} \to \mathbb{R}^d$
2: **for** each gradient step **do**
3:     Sample batch $\{(s, a)\} \sim \mathcal{D}$
4:     $a_{\text{base}} \leftarrow \pi_{\text{base}}(s), \Delta a \leftarrow a - a_{\text{base}}$
5:     $\tilde{A} \leftarrow \big(Q_{\text{rob}}(s, a) - Q_{\text{rob}}(s, a_{\text{base}})\big)/\sigma_Q(s)$
6:     $w \leftarrow \exp(\tilde{A}/T)$
7:     $\mathcal{L}_{\text{fit}} \leftarrow \mathbb{E}\big[w \cdot \|\pi_{\text{res}}(s, a_{\text{base}}) - \Delta a\|_2^2\big]$
8:     $a_{\text{synth}} \leftarrow a_{\text{base}} + \pi_{\text{res}}(s, a_{\text{base}})$
9:     $\mathcal{L}_{\text{guide}} \leftarrow -\lambda_g \cdot \mathbb{E}\big[Q_{\text{rob}}(s, a_{\text{synth}})\big]$ {Gradient ascent}
10:     Update $\pi_{\text{res}}$ via $\nabla_\psi\big(\mathcal{L}_{\text{fit}} + \mathcal{L}_{\text{guide}}\big)$
11: **end for**
12: **return** $\pi_{\text{res}}$

---

---

**Algorithm 3** SPAR-PLAS

---

**Require:** $\pi_{\text{base}}$, dataset $\mathcal{D}$, critics $\{Q_{\theta_i}\}$
**Ensure:** Latent policy $\pi_z$ and frozen CVAE decoder $\pi_{\text{res}}$
1: *// Phase 1: Pre-train CVAE via weighted ELBO*
2: Initialize CVAE encoder $q_\phi(z|s, a, \Delta a)$ and decoder $p_\psi(\Delta a|s, a, z)$
3: **for** each gradient step in Phase 1 **do**
4:     Sample $\{(s, a)\} \sim \mathcal{D}$, compute $a_{\text{base}} = \pi_{\text{base}}(s)$, $\Delta a_{\text{gt}} = a - a_{\text{base}}$
5:     Compute weight $w = \exp(\tilde{A}/T) \cdot \mathbb{I}\{\tilde{A} > 0\}$ with $\tilde{A} = (Q_{\text{rob}}(s, a) - Q_{\text{rob}}(s, a_{\text{base}}))/\sigma_Q$
6:     Update CVAE via weighted ELBO: $\mathcal{L}_{\text{fit}} = \mathbb{E}[w \cdot \|\pi_{\text{res}}(s, a_{\text{base}}, z) - \Delta a_{\text{gt}}\|_2^2] + \beta \cdot \text{KL}(q_\phi\|p(z))$
7: **end for**
8: **Freeze** CVAE decoder parameters (stop-gradient)
9: *// Phase 2: Train latent policy via gradient guidance (frozen CVAE)*
10: Initialize deterministic latent policy $\pi_z : \mathcal{S} \times \mathcal{A} \to \mathcal{Z}$
11: **for** each gradient step in Phase 2 **do**
12:     Sample $\{s\} \sim \mathcal{D}$, $a_{\text{base}} \leftarrow \pi_{\text{base}}(s)$
13:     $z \leftarrow \pi_z(s, a_{\text{base}})$ {Deterministic latent}
14:     $\Delta a \leftarrow \pi_{\text{res}}(s, a_{\text{base}}, z)$ {Frozen decoder}
15:     $a_{\text{synth}} \leftarrow \text{clip}(a_{\text{base}} + \Delta a, -1, 1)$
16:     $\mathcal{L}_{\text{guide}} \leftarrow -\lambda_g \cdot Q_{\text{rob}}(s, a_{\text{synth}})$ {Gradient ascent}
17:     Update $\pi_z$ via $\nabla_z \mathcal{L}_{\text{guide}}$ {CVAE remains frozen}
18: **end for**
19: **return** $\pi_z$, $\pi_{\text{res}}$

---

**Algorithm 4** SPAR-PROJ

---

**Require:** $\pi_{\text{base}}$, CVAE $\pi_{\text{res}}$, target CVAE $\pi'_{\text{res}}$, critics
**Ensure:** Projection-guided residual policy
1: Initialize $\pi_{\text{res}}, \pi'_{\text{res}} \leftarrow \pi_{\text{res}}$
2: **for** each gradient step **do**
3:     Sample batch $\{s\} \sim \mathcal{D}$, $a_{\text{base}} \leftarrow \pi_{\text{base}}(s)$
4:     *// Step 1: Value-weighted sampling*
5:     Sample $\{z_k\}_{k=1}^K \sim \mathcal{N}(0, I)$
6:     $\Delta a_k \leftarrow \pi'_{\text{res}}(s, a_{\text{base}}, z_k)$ {Target decoder}
7:     $a_k \leftarrow a_{\text{base}} + \Delta a_k$
8:     $\tilde{A}_k \leftarrow (Q_{\text{rob}}(s, a_k) - Q_{\text{rob}}(s, a_{\text{base}}))/\sigma_Q(s)$
9:     $\omega_k \leftarrow \frac{\exp(\tilde{A}_k/T)}{\sum_j \exp(\tilde{A}_j/T)}$
10:     *// Step 2: Projection update*
11:     $\mathcal{L}_{\text{fit}} \leftarrow \mathbb{E}[\sum_k \omega_k \|\pi_{\text{res}}(s, a_{\text{base}}, z_k) - \Delta a_k\|_2^2] + \beta \cdot \text{KL}(q_\phi\|p(z))$ {Weighted ELBO}
12:     $\mathcal{L}_{\text{guide}} \leftarrow \mathbb{E}[\sum_k \omega_k \|\pi_{\text{res}}(s, a_{\text{base}}, z_k) - \Delta a_k\|_2^2]$ {Value-weighted regression}
13:     Update $\pi_{\text{res}}$ via $\nabla_\psi(\mathcal{L}_{\text{fit}} + \mathcal{L}_{\text{guide}})$
14:     $\pi'_{\text{res}} \leftarrow (1 - \tau)\pi'_{\text{res}} + \tau\pi_{\text{res}}$ {Polyak averaging}
15: **end for**
16: **return** $\pi_{\text{res}}$

---

---

**Algorithm 5** Stage III: Deployment-Time Rectification

---

**Require:** State $s$, $\pi_{\text{base}}$, $\pi_{\text{res}}$, $\eta_{\text{abs}} = 10^{-4}$, $\eta_{\text{rel}} = 0.01$
**Ensure:** Action $a_{\text{spar}}$
 1: $a_{\text{base}} \leftarrow \pi_{\text{base}}(s)$
 2: $Q_{\text{base}} \leftarrow Q_{\text{rob}}(s, a_{\text{base}})$
 3: **for** $k = 1$ to $K$ **do**
 4:     Get $\Delta a_k$ (det. or sampled $z_k$)
 5:     $a_k \leftarrow a_{\text{base}} + \Delta a_k$
 6:     $\Delta Q_k \leftarrow Q_{\text{rob}}(s, a_k) - Q_{\text{base}}$
 7:     $R_k \leftarrow \Delta Q_k / (|Q_{\text{base}}| + \epsilon)$
 8: **end for**
 9: $k^* \leftarrow \arg\max_k \Delta Q_k$
10: **if** $\Delta Q_{k^*} > \eta_{\text{abs}}$ **and** $R_{k^*} > \eta_{\text{rel}}$ **then**
11:     **return** $a_{k^*}$
12: **else**
13:     **return** $a_{\text{base}}$
14: **end if**

---

# C. Experiments

## C.1. Hyperparameter Configuration

This section provides a comprehensive specification of hyperparameters used in SPAR's three-stage pipeline. All experiments use JAX/Flax implementation with Adam optimizer and fixed random seeds $\{0, 42, 123\}$.

### C.1.1. GLOBAL HYPERPARAMETERS

Table 7 summarizes architecture-agnostic settings shared across all stages and environments.

### C.1.2. STAGE I: CONSERVATIVE ANCHOR

Stage I trains a pure behavior cloning policy $\pi_{\text{base}}$ and conservative critic ensemble using IQL. Critical configuration differences:

- **Expectile parameter** $\tau$: Set to 0.5 for most environments to approximate conditional mean of behavior policy. For sparse-reward AntMaze tasks, increased to 0.9 to restore critic discriminability (as noted in Sec. 4.2 of main text).

- **Policy type**: Deterministic for MuJoCo locomotion (`base_policy_deterministic=True`), stochastic Gaussian for Adroit/AntMaze (`False`) with no advantage scaling at all.

- **IQL-specific**: $\beta = 3.0$ (advantage scaling), `iql_tau` $= 0.7$ (value loss expectile).

Learning rates: $3 \times 10^{-4}$ for all networks (actor, critic, value function). Target network update rate $\tau_{\text{polyak}} = 0.005$.

### C.1.3. STAGE II: RESIDUAL POLICY LEARNING

Architecture selection follows residual topology analysis (Sec. 4.2):

- **SPAR-MLP**: Used for unimodal residuals (MuJoCo Medium-Replay). Direct regression with gradient-based guidance:

$$\mathcal{L}_{\text{total}} = \mathbb{E}[w(s, a) \cdot \|\pi_{\text{res}} - \Delta a\|_2^2] - \lambda_g \mathbb{E}[Q_{\text{rob}}(s, a_{\text{synth}})] \tag{38}$$

- **SPAR-PROJ**: Used for multimodal/narrow residuals (MuJoCo Medium-Expert, Adroit, AntMaze). CVAE with latent self-imitation (Algorithm 4). Key parameters:
  - Projection period: 10 steps (How often to perform latent self-imitation)
  - Target network EMA: $\tau_{\text{target}} = 0.005$

*Table 7.* Global hyperparameters shared across all D4RL benchmarks.

| Component | Value |
|---|---|
| *Optimization* | |
| Batch size | 256 |
| Discount factor ($\gamma$) | 0.99 |
| Optimizer | Adam |
| Gradient clipping | 1.0 (global norm) |
| Max consecutive errors | 10 |
| *Network Architecture* | |
| Hidden dimension (MLP/CVAE) | 256 |
| Number of hidden layers | 3 (encoder/decoder) |
| CVAE latent dimension ($d_z$) | 16 |
| CVAE KL weight ($\beta_{\mathrm{KL}}$) | 0.5 |
| CVAE reconstruction weight | 1.0 |
| Latent policy hidden dims | (256, 256) |
| MLP residual hidden dims | (256, 256) |
| Actor max action | 1.0 (normalized action space) |
| *Ensemble & Uncertainty* | |
| Total critics | 10 |
| Data critics subset ($M_{\mathrm{data}}$) | 4 |
| Guide critics subset ($M_{\mathrm{guide}}$) | 4 |
| Rectification critics ($M_{\mathrm{rect}}$) | 4 |
| Uncertainty penalty ($\lambda_u$) | Task-dependent |
| *Training Schedule* | |
| Stage I steps | $1 \times 10^6$ |
| Stage II steps | $1 \times 10^6$ |
| Evaluation frequency | 50,000 steps |
| Logging frequency | 1,000 steps |
| *Normalization* | |
| State normalization | Enabled (zero-mean, unit-var) |
| Reward normalization | Disabled (except antmaze) |

- Candidate samples: $K = 64$
- Relative gain computation: enabled

Table 8 summarizes SPAR's best settings of dynamic hyperparameters across D4RL benchmarks.

*Table 8.* Hyperparameters for Best SPAR across 10 D4RL benchmarks. Organized by the three-stage structure.

| Domain | Task | Stage I | Stage II (Residual Learning) | | | | | Stage III |
|---|---|---|---|---|---|---|---|---|
| | | $\tau$ | Arch | $\lambda_u$ | $\lambda_g$ | Filter | $T$ | $\eta$ |
| MuJoCo MR | HalfCheetah | 0.5 | MLP | 0.5 | 0.5 | Hard | 1.0 | 0.01 |
| | Hopper | 0.5 | MLP | 0.5 | 0.5 | Hard | 1.0 | 0.01 |
| | Walker2d | 0.5 | MLP | 0.5 | 2.0 | Hard | 1.0 | 0.01 |
| MuJoCo ME | HalfCheetah | 0.5 | PROJ | 0.5 | 0.5 | Hard | 0.3 | 0.01 |
| | Hopper | 0.5 | PROJ | 2 | 0.5 | Hard | 0.3 | 0.01 |
| | Walker2d | 0.5 | PROJ | 0.5 | 2.0 | Hard | 0.3 | 0.01 |
| Adroit | Pen-Cloned | 0.5 | PROJ | 0.5 | 0.5 | Soft | 0.3 | 0.01 |
| | Pen-Human | 0.5 | PROJ | 0.5 | 0.5 | Soft | 0.3 | 0.01 |
| AntMaze | Umaze-Div | 0.9 | PROJ | 0.5 | 2.0 | Soft | 0.3 | 0.01 |
| | Large-Div | 0.9 | PROJ | 0.5 | 0.5 | Soft | 0.3 | 0.01 |

## C.1.4. STAGE III: DEPLOYMENT-TIME RECTIFICATION

Value-gated action selection with dual-threshold mechanism:

- Candidate sampling: $K = 10$ latent samples

- Absolute threshold: $\eta_{\text{abs}} = 10^{-4}$

- Relative threshold: $\eta_{\text{rel}} = 0.01$

- Dual-threshold logic: Accept residual iff $\Delta Q > \eta_{\text{abs}} \wedge \frac{\Delta Q}{|Q_{\text{base}}| + \epsilon} > \eta_{\text{rel}}$

These thresholds are shared across tasks and serve as a conservative deployment-time safety filter with fixed settings.

### C.2. Additional Experiments

This section reports additional baseline comparisons and diagnostic analyses under the same D4RL evaluation protocol and score normalization as the main experiments.

#### C.2.1. RECENT BASELINES

Table 9 compares SPAR with recent diffusion-, flow-, and in-support baselines on representative tasks.

*Table 9.* Comparison with recent offline RL baselines. Higher is better.

| Task | SPAR | DAC | FQL | QIPO |
|------|------|------|------|------|
| HP-MR | **101.9** | 99.8 | 64.8 | 101.2 |
| HC-ME | 97.0 | 47.6 | **106.1** | 94.0 |
| Pen-CL | 76.2 | **78.7** | 52.7 | 35.0 |
| AM-LD | **71.0** | 60.0 | 40.0 | 40.0 |

#### C.2.2. STAGE III THRESHOLD SENSITIVITY

Table 10 evaluates the two deployment-time gating thresholds. The shared default $(\eta_{\text{abs}}, \eta_{\text{rel}}) = (10^{-4}, 0.01)$ performs best overall, while extreme settings predictably degrade performance.

*Table 10.* Sensitivity of Stage III thresholds.

| $\eta_{\text{abs}}$ | $\eta_{\text{rel}}$ | **Pen-CL** | **HP-MR** | **HC-ME** | **AM-LD** |
|------|------|------|------|------|------|
| $10^9$ | $10^9$ | 50.4 | 42.9 | 63.1 | 20.0 |
| $-10^9$ | $-10^9$ | 61.6 | 49.5 | 96.1 | 60.0 |
| $-10^9$ | 0.01 | 68.1 | 95.2 | 92.0 | 55.0 |
| $10^{-4}$ | $-10^9$ | 67.2 | 78.2 | 96.4 | 55.0 |
| $10^{-4}$ | 0.01 | **76.2** | **101.9** | **97.0** | **71.0** |

#### C.2.3. INFERENCE LATENCY AND MEMORY

Table 11 reports average inference latency and peak memory usage measured with batch size 1 and averaged over 1,500 inference steps. SPAR-PROJ requires one residual generation pass and a small candidate set at Stage III, avoiding iterative denoising.

*Table 11.* Inference latency and peak memory usage.

| Task | SPAR-PROJ ms | SPAR-PROJ MB | DAC ms | DAC MB |
|------|------|------|------|------|
| Pen-CL | 0.394 | 1412.52 | 0.577 | 1608.66 |
| HP-MR | 0.366 | 1389.48 | 0.589 | 1590.12 |
| HC-ME | 0.409 | 1402.66 | 0.616 | 1592.66 |
| AM-LD | 0.403 | 1396.74 | 0.535 | 1592.79 |

### C.2.4. FLOW-BASED GENERATOR VARIANTS

Table 12 tests whether simply increasing generator expressiveness explains SPAR's gains. The results suggest that the anchored residual formulation and support-preserving improvement mechanism are the primary contributors.

*Table 12.* Flow-based base and residual variants under matched settings.

| Task | MLP base | MLP base + flow | Flow base | Flow base + flow | SPAR |
|---|---|---|---|---|---|
| HP-MR | 42.9 | 97.5 | 31.4 | 96.2 | **101.9** |
| HC-ME | 63.1 | 75.5 | 69.3 | 77.8 | **97.0** |
| Pen-CL | 50.4 | 68.2 | 64.6 | 68.5 | **76.2** |
| AM-LD | 20.0 | 5.0 | 10.0 | 10.0 | **71.0** |

*Table 13.* SPAR residual learning on top of a flow-matching base policy.

| Task | Flow-base Score | Residual Policy | SPAR Score |
|---|---|---|---|
| HP-MR | 31.4 | SPAR-MLP | 76.5 |
| HC-ME | 69.3 | SPAR-PROJ | 94.9 |
| Pen-CL | 64.6 | SPAR-PROJ | 65.3 |
| AM-LD | 10.0 | SPAR-PROJ | 40.0 |

### C.2.5. MUJOCO MEDIUM DATASETS

Table 14 reports additional MuJoCo medium results. SPAR-MLP is effective on these more nearly unimodal residual distributions.

*Table 14.* Additional MuJoCo medium results.

| Task | MLP-BASE | SPAR-MLP | SPAR-PROJ |
|---|---|---|---|
| HP-M | 53.6 | **91.2** | 87.4 |
| HC-M | 42.3 | **55.0** | 47.8 |
| WK-M | 76.8 | **88.1** | 82.9 |

### C.2.6. ARCHITECTURE SELECTION

For architecture selection, we compute empirical residuals $\Delta a = a - \pi_{\text{base}}(s)$ on the dataset. A compact, roughly unimodal residual distribution suggests that SPAR-MLP is sufficient, while fragmented, multi-cluster, narrow, or sparse residual geometry suggests SPAR-PROJ. A lightweight diagnostic can use covariance-spectrum concentration, nearest-neighbor tail distances, and density-based clustering that leaves the number of clusters open. A concentrated spectrum with one dense cluster indicates compact residuals; disconnected clusters, high tail distances, or many small clusters indicate fragmented support. These diagnostics are computed in residual/action space, and low-dimensional plots are used only for visualization. In practice, SPAR-PROJ is a robust default because its generative residual model can represent both unimodal and multimodal residuals.

### C.3. Full Visualization of Residuals

Fig. 5 shows the visualization of the $(s, a)$ distribution across 10 tasks.

### C.4. Full Visualization of Training Curves

Fig. 6 shows training curves for the highest-scoring settings across 10 tasks, averaged over three random seeds (0, 42, 123).

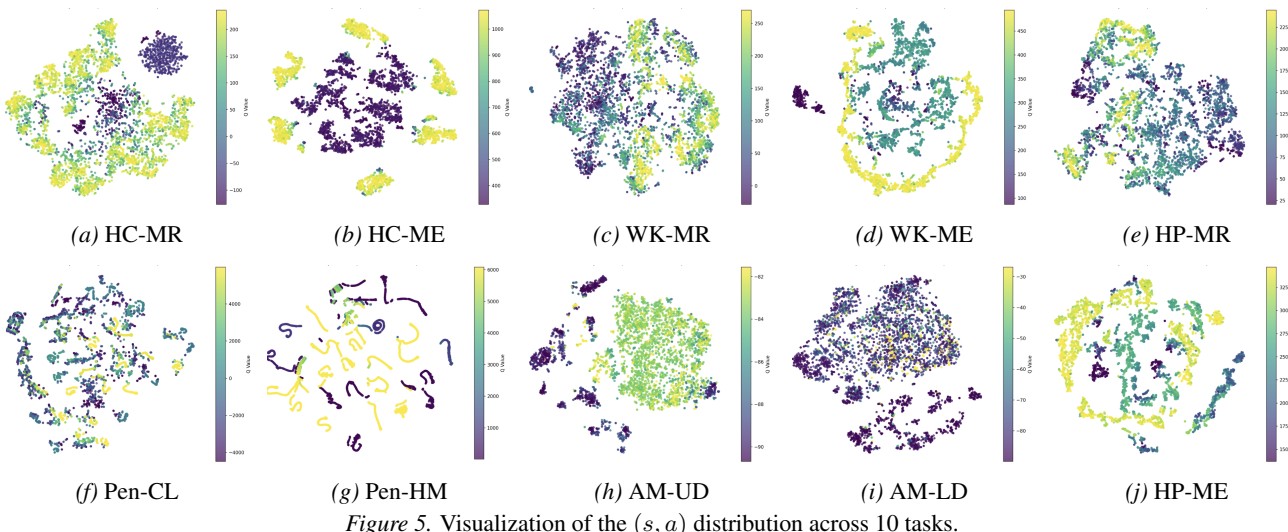

*(a)* HC-MR      *(b)* HC-ME      *(c)* WK-MR      *(d)* WK-ME      *(e)* HP-MR

*(f)* Pen-CL      *(g)* Pen-HM      *(h)* AM-UD      *(i)* AM-LD      *(j)* HP-ME

*Figure 5.* Visualization of the $(s, a)$ distribution across 10 tasks.

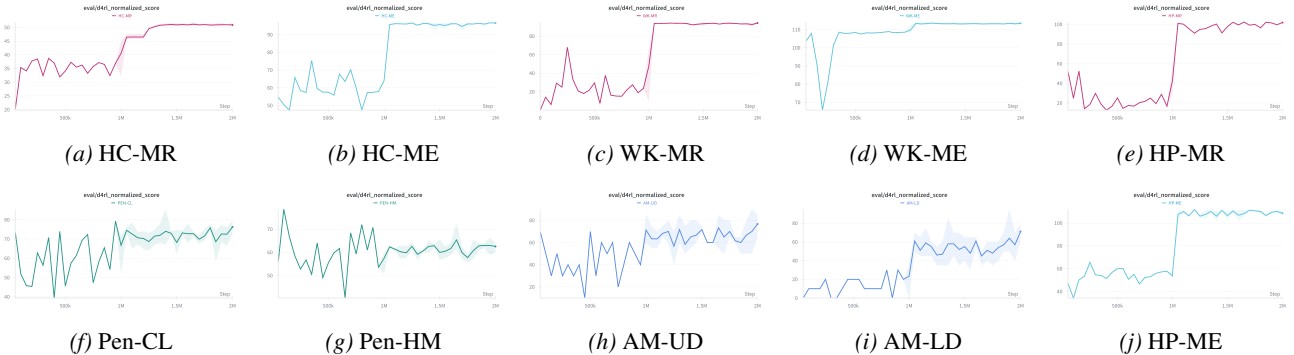

*(a)* HC-MR      *(b)* HC-ME      *(c)* WK-MR      *(d)* WK-ME      *(e)* HP-MR

*(f)* Pen-CL      *(g)* Pen-HM      *(h)* AM-UD      *(i)* AM-LD      *(j)* HP-ME

*Figure 6.* Visualization of the training curves for the highest-scoring settings across 10 tasks, averaged over three random seeds (0, 42, 123). The training step length for Stage I is 1M, and the training step length for Stage II is also 1M, starting from step = 1M.

