# OpenReview forum: "SPAR: Support-Preserving Action Rectification"
_ICML.cc/2026/Conference — ICML 2026 regular_

### Official Review · Reviewer_9bPY · 2026-03-11

**Soundness:** 2
**Presentation:** 3
**Significance:** 3
**Originality:** 3
**Overall Recommendation:** 4
**Confidence:** 2

**Summary:**

This paper proposes SPAR, an offline RL framework that improves a frozen behavior cloning policy through residual learning while preserving dataset support. The key idea is to restrict policy improvement to a local residual space around the BC action, which reduces the effective search space and mitigates off-manifold errors. For multimodal settings, the method introduces latent self-imitation, which replaces direct value-gradient updates with latent sampling and value-weighted regression. The paper provides theoretical motivation for this design and demonstrates strong empirical performance on D4RL benchmarks, with SPAR-MLP working well in unimodal regimes and SPAR-PROJ showing advantages in multimodal and sparse-reward tasks.

**Compliance With Llm Reviewing Policy:**

Affirmed.

**Final Justification:**

The authors have addressed most of my concerns, so I decided to keep my score at 4.

**Key Questions For Authors:**

* Q1: How should practitioners determine whether a given dataset is “unimodal” enough for SPAR-MLP or “multimodal” enough for SPAR-PROJ?

The paper’s mechanism analysis suggests that the choice of residual policy class is crucial, with SPAR-MLP working well in MuJoCo medium-replay but failing badly in multimodal regimes, while SPAR-PROJ is more effective there. It would be helpful to know whether the authors have a quantitative diagnostic, beyond post hoc visualization, for selecting between the two variants.

* Q2: Have the authors considered using a diffusion model instead of the BC policy or the CVAE-based residual model, and if so, would this further improve performance?

Since one of the main motivations of the paper is handling multimodal action distributions, it would be interesting to understand whether replacing the BC anchor with a diffusion-based behavior model, or replacing the CVAE-style SPAR-PROJ residual policy with a diffusion model, could provide additional benefits.

**Limitations:**

yes

**Strengths And Weaknesses:**

$\textbf{Strengths}$

The paper presents a novel and well-motivated offline RL framework that decouples policy fitting and policy improvement through a frozen BC anchor and residual correction. This support-preserving formulation is conceptually interesting, and the latent self-imitation mechanism provides a neat alternative to direct critic-gradient optimization for multimodal settings. Another major strength is the theoretical motivation: the paper explains the method through search-space contraction and the conflict between fitting and optimization, which gives a principled justification beyond purely empirical design.

$\textbf{Weaknesses}$

A main weakness is the lack of comparison with recent strong diffusion-based offline RL methods, which is especially important since multimodal action modeling is one of the paper’s main motivations. In addition, the method seems sensitive to the residual geometry of the dataset: SPAR-MLP works well in unimodal settings but can fail badly in multimodal ones, while SPAR-PROJ may be unnecessarily conservative in simpler regimes.

---

> ### Author Rebuttal · Authors · 2026-03-30
>
> We sincerely thank the reviewer for recognizing the conceptual novelty of our support-preserving formulation, the latent self-imitation mechanism, and the theoretical motivation behind our design. We address the concerns and questions below.
>
> **W1: Missing Recent Diffusion-/Flow-Based Baselines:**
> We agree that comparisons against stronger recent baselines are important, especially because handling multimodal action distributions is one of the key motivations of SPAR. To strengthen empirical positioning, we add comparisons against a representative subset of recent diffusion-/flow-style offline RL methods. [T1] reports results under the same D4RL evaluation protocol and score normalization as our main results. The main takeaway is that SPAR remains competitive or superior on most of these representative tasks, while the added comparisons better situate our method among current generative-policy baselines. We will also expand the related-work discussion accordingly in the revision.
>
> |Env|SPAR|DAC|FQL|QIPO|
> |:-:|:-:|:-:|:-:|:-:|
> |hp-mr|101.9|99.8|64.8|101.2|
> |hc-me|97.0|47.6|106.1|94.0|
> |pen-cl|76.2|78.7|52.7|35.0|
> |am-ld|70.0|60.0|40.0|40.0|
>
> **W2&Q1: Architecture Selection and Practical Applicability:**
> We clarify that SPAR-MLP and SPAR-PROJ are two instantiations of the same framework, intended for different residual topologies rather than different algorithms. Importantly, practitioners do not need visualization tools such as t-SNE to choose between them. A lightweight data-driven criterion is to compute empirical residuals $Δa=a−π_{base}(s)$ on the dataset and examine their geometry via unsupervised clustering without assuming the number of clusters a priori. A roughly unimodal, compact residual distribution suggests SPAR-MLP is sufficient, whereas a fragmented or multi-cluster residual structure indicates SPAR-PROJ is more appropriate. In practice, we recommend SPAR-PROJ as the default setting, since its generative parameterization can handle both unimodal and multimodal residuals, whereas a deterministic MLP is more prone to mean-seeking under multimodality.
>
> **Q2: Would Replacing the BC Anchor or the CVAE Residual Model with a Stronger Generative Model Further Improve Performance:**
> We have considered replacing either the BC anchor or the residual model with a stronger generative module. In SPAR, however, the generative module is not required to model the full action distribution from scratch; it only needs to capture the local residual distribution around a frozen anchor. This substantially reduces the modeling burden relative to global policy generation, which is why a CVAE is already a natural choice within our framework.
>
> To test whether simply increasing generator expressiveness would improve performance, we conducted preliminary experiments with stronger flow-based base/residual variants as a first proxy. [T2] reports these results under matched settings. The key observation is that more expressive generators do not lead to clear or consistent gains over the default SPAR design. This suggests that SPAR’s main advantage comes from the localized residual formulation and support-preserving improvement mechanism, rather than from generator expressiveness alone.
>
> |Task|MLP base|MLP base + flow policy|Flow base|Flow base + flow policy|SPAR|
> |:-:|:-:|:-:|:-:|:-:|:-:|
> |hp-mr|42.9|97.5|31.4|96.2|101.9|
> |hc-me|63.1|75.5|69.3|77.8|97.0|
> |pen-cl|50.4|68.2|64.6|68.5|76.2|
> |am-ld|20.0|5.0|10.0|10.0|70.0|
>
> We will clarify that our use of a CVAE is motivated by the localized nature of residual modeling and the empirical trade-off we observed, rather than by any claim of universal superiority over diffusion-based or flow-based generators.

---

> > ### Author Rebuttal · Reviewer_9bPY · 2026-04-03
> >
> > Thank you for addressing my concerns. I will maintain the current score.

---

> > > ### Author Response · Authors · 2026-04-07
> > >
> > > Thank you for confirming resolution. We will incorporate all discussed additions in the rivision version.

---

### Official Review · Reviewer_g7LT · 2026-03-12

**Soundness:** 2
**Presentation:** 3
**Significance:** 2
**Originality:** 3
**Overall Recommendation:** 4
**Confidence:** 4

**Summary:**

This paper proposes SPAR, a residual-policy framework for offline RL that decouples behavior fitting from value-driven improvement. A frozen behavior cloning base policy provides an anchor, and a residual policy performs local rectification within a contracted residual space, with a latent self-imitation mechanism that guides improvements by value-weighted sampling on the residual manifold rather than direct gradient ascent. The authors provide geometric arguments for why the sampling-based residual updates avoid off-manifold drift and present D4RL experiments showing strong performance across MuJoCo, AntMaze, and Adroit tasks.

**Compliance With Llm Reviewing Policy:**

Affirmed.

**Final Justification:**

The rebuttal addressed most of my concerns. Considering both the paper and the rebuttal, I lean toward acceptance, but I would not object if the paper is rejected.

**Key Questions For Authors:**

1. How can practitioners decide between SPAR-MLP and SPAR-PROJ without visualizing residual distributions (e.g., t-SNE)? Is there an automatic or data-driven criterion for selecting the appropriate architecture?

2. How sensitive are results to the gating thresholds (ηabs, ηrel) in Stage III? Did you tune these per task, and if so, how was fairness enforced relative to baselines?

3. Can you provide stronger OOD diagnostics (e.g., behavior-density estimates or distance to a behavior manifold) to corroborate the claim that SPAR-PROJ reduces off-support drift, beyond the gradient-conflict metrics?

4. Several recent and closely related baselines (DTQL, FAC, FQL, OFQL [4], ...), as well as results on the MuJoCo medium datasets, are missing. Can the authors add these comparisons or discuss in the related work why they are out of scope? If computational resources are a limitation, reporting results on a focused subset would still be informative.

[4] One-Step Flow Q-Learning: Addressing the Diffusion Policy Bottleneck in Offline Reinforcement Learning

**Limitations:**

yes

**Strengths And Weaknesses:**

# Strength:
- The paper proposes SPAR, which reformulates offline policy improvement as residual rectification around a frozen behavior cloning policy. This decoupling between behavior fitting and value-driven improvement provides a principled way to constrain optimization within the behavior support while still enabling policy improvement. This is a conceptually novel residual-policy formulation.
- The proposed latent self-imitation (LSI) replaces gradient-based actor optimization with value-weighted sampling in a learned residual manifold. This is an interesting alternative to gradient ascent in offline RL and aims to mitigate off-support drift caused by critic errors.
- The method is evaluated on several D4RL benchmarks (MuJoCo, AntMaze, and Adroit). The ablation studies on guidance weight, uncertainty weighting, and filtering strategies provide useful insight into the algorithm’s behavior.

# Weaknesses:

- The theoretical analysis mainly focuses on search-space contraction and geometric drift arguments, but it does not provide guarantees related to offline RL stability, Bellman error propagation, or policy performance under distribution shift.

- Missing comparisons to several very recent and conceptually close SOTA methods that also decouple or regularize improvement  (e.g., DTQL [1], FAC [2], FQL [3]...). Given the paper’s framing and claims, these comparisons are important for positioning and fairness.

- The method requires choosing between two residual architectures (MLP vs CVAE) depending on the residual topology of each task domain. This reliance on prior knowledge reduces the plug-and-play nature of the approach and raises questions about general applicability

[1] Diffusion Policies creating a Trust Region for Offline Reinforcement Learning \
[2] Flow Actor-Critic for Offline Reinforcement Learning \
[3] Flow Q Learning

---

> ### Author Rebuttal · Authors · 2026-03-30
>
> We sincerely thank the reviewer for recognizing the conceptual novelty of our work. We address the concerns and questions below.
>
> **W1: Theoretical Scope:**
> We agree that our current theory is narrower in scope. Its goal is mechanism-level: to justify why SPAR reduces manifold-normal drift relative to direct value-gradient optimization by (i) contracting the search space around a frozen BC anchor and (ii) restricting improvement to residual support. We will clarify this scope explicitly in the revision and discuss stronger offline-RL guarantees as an important future direction.
>
> **W2&Q4: Missing Recent Baselines and MuJoCo Medium Datasets:**
> We thank the reviewer for highlighting recent diffusion/flow-style methods. We agree these are important for positioning. In the revision, we will expand the related-work discussion accordingly and, within the rebuttal budget, add comparisons to a representative subset of recent baselines, as shown in [T1]:
>
> |Env|SPAR|DAC|FQL|QIPO|
> |:-:|:-:|:-:|:-:|:-:|
> |hp-mr|101.9|99.8|64.8|101.2|
> |hc-me|97.0|47.6|106.1|94.0|
> |pen-cl|76.2|78.7|52.7|35.0|
> |am-ld|70.0|60.0|40.0|40.0|
>
> SPAR remains competitive or superior on most of these representative tasks, while the added comparisons improve empirical positioning.
>
>  We will also include results on the MuJoCo medium datasets in the appendix, as shown in [T2], SPAR-MLP performs better on these more nearly unimodal datasets.
>
> |Env|MLP-BASE|SPAR-MLP|SPAR-PROJ|
> |:-:|:-:|:-:|:-:|
> |hp-m|53.6|91.2|87.4|
> |hc-m|42.3|55.0|47.8|
> |wk-m|76.8|88.1|82.9|
>
>
> **W3&Q1: Architecture Selection and General Applicability:**
> We clarify that SPAR-MLP and SPAR-PROJ are two instantiations of the same framework, intended for different residual topologies rather than different algorithms. Importantly, practitioners do not need visualization tools such as t-SNE to choose between them. A lightweight data-driven criterion is to compute empirical residuals $Δa=a−π_{base}(s)$ on the dataset and examine their geometry via unsupervised clustering without assuming the number of clusters a priori. A roughly unimodal, compact residual distribution suggests SPAR-MLP is sufficient, whereas a fragmented or multi-cluster residual structure indicates SPAR-PROJ is more appropriate. In practice, we recommend SPAR-PROJ as the default setting, since its generative parameterization can handle both unimodal and multimodal residuals, whereas a deterministic MLP is more prone to mean-seeking under multimodality.
>
> **Q2: Sensitivity to Stage III Gating Thresholds:**
> We emphasize that the Stage III thresholds were not heavily tuned on a per-task basis($\eta_{abs}=10^{-4}, \eta_{rel}=0.01$). Instead, we used a shared default configuration across most domains, so Stage III acts as a conservative safety filter rather than a task-specific performance knob. This avoids giving SPAR extra task-specific tuning advantages relative to baselines. To directly address this point, [T3] ablates the thresholds under the same training/evaluation setup. The result is consistent: the shared default performs best overall, while extreme settings predictably degrade performance.
>
> |$\eta_{abs}$|$\eta_{rel}$|pen-cl|hp-mr|hc-me|am-ld|
> |:-:|:-:|:-:|:-:|:-:|:-:|
> |1e9|1e9|50.4|42.9|63.1|20.0|
> |-1e9|-1e9|61.6|49.5|96.1|60.0|
> |-1e9|0.01|68.1|95.2|92.0|55.0|
> |1e-4|-1e9|67.2|78.2|96.4|55.0|
> |1e-4|0.01|76.2|101.9|97.0|70.0|
>
> **Q3: Stronger OOD Diagnostics**
> We agree that stronger direct evidence of support preservation would further strengthen the paper. In the revision, we will add a compact dataset-reference support diagnostic with both quantitative and visual evidence, as shown in https://anonymous.4open.science/r/ICML2026-31415-Rebuttal-277D/fig1.png. We compare dataset actions with actions generated by SPAR-PROJ and SPAR-PLAS. Quantitatively, we will report a kNN-based support-distance metric in the original action space, measuring how far policy actions deviate from the empirical dataset support; lower values indicate closer alignment with the behavior manifold. Visually, we will provide a concise shared-space boundary plot. Together, these diagnostics more directly test whether SPAR-PROJ stays closer to dataset support than SPAR-PLAS.

---

> > ### Author Rebuttal · Reviewer_g7LT · 2026-04-03
> >
> > Thank you for addressing my concerns. I have increased my score.

---

> > > ### Author Response · Authors · 2026-04-07
> > >
> > > Thank you for confirming resolution. We will incorporate all discussed additions in the rivision version.

---

### Official Review · Reviewer_i7xa · 2026-03-13

**Soundness:** 3
**Presentation:** 3
**Significance:** 2
**Originality:** 2
**Overall Recommendation:** 4
**Confidence:** 4

**Summary:**

In gradient-based offline RL algorithms, conflicts between data fitting and value maximization may drive the learned policy off the data manifold. To address this issue, this paper proposes SPAR, a three-stage pipeline that aims to optimize policies within the support of the behavior policy. In the first stage, SPAR trains a base policy via behavior cloning. In the second stage, a residual policy is learned through weighted regression and self-imitation learning. In the third stage, the residual is set to zero when the value improvement is small, which aims to enhance safety. Experiments on the D4RL benchmarks demonstrate that SPAR alleviates the gradient conflicts between data fitting and value maximization, and outperforms several gradient-based and in-support learning baselines on most tasks.

**Compliance With Llm Reviewing Policy:**

Affirmed.

**Final Justification:**

The authors' response addressed my concerns. I have decided to give a positive score.

**Key Questions For Authors:**

- In Line 176, the paper states that the residual policy is trained using a weighted ELBO objective. Since ELBO consists of a reconstruction term and a KL divergence term, why is the weight applied only to the reconstruction term rather than to both terms?


- SPAR_proj adopts a CVAE to model the residual policy. Could the authors clarify why CVAE is chosen instead of more expressive generative models, such as diffusion models, which have recently shown strong performance in offline RL?


- The base policy is implemented as either a deterministic policy or a Gaussian policy. Considering that offline datasets can be highly multimodal, such simple policy parameterizations may result in a relatively weak base policy. Could the authors discuss whether this might limit the effectiveness of the subsequent residual policy learning?


[1] Fang L, Liu R, Zhang J, et al. Diffusion Actor-Critic: Formulating Constrained Policy Iteration as Diffusion Noise Regression for Offline Reinforcement Learning. International Conference on Learning Representations. 2025.

[2] Park S, Li Q, Levine S. Flow q-learning [C]. International Conference on Machine Learning. 2025.

[3] Zhang S, Zhang W, Gu Q. Energy-Weighted Flow Matching for Offline Reinforcement Learning. International Conference on Learning Representations. 2025.

**Limitations:**

Yes.

**Strengths And Weaknesses:**

### Strengths

- The paper addresses an important challenge in offline RL, namely the conflict between data fitting and value maximization.

- The proposed three-stage pipeline provides an intuitive way to restrict policy updates within the support of the behavior policy.

### Weaknesses

- The paper claims that, unlike gradient-based methods, SPAR can prevent policy updates from drifting off the support of the dataset. To better support this claim, it would be helpful to include visualizations comparing the actions in the dataset and those generated by the learned policy, which could provide more direct evidence of whether the learned policy remains within the data support.

- SPAR consists of three key components:

  1. learning a residual policy to reduce the search space
  2. self-imitation learning to explore potential high-value actions
  3. value-gated rectification at test time to improve robustness.

  However, the paper lacks ablation studies on these components. Including experiments where each component is removed would help clarify their individual contributions and demonstrate the necessity of the full pipeline.

- The adopted baselines appear somewhat outdated. It would strengthen the empirical evaluation to include more recent offline RL methods, for example:

  Gradient-based methods: DAC [1], FQL [2],
  In-support learning: QIPO [3]

- Minor issues:

  - There are display errors in the references cited in Line 46 and Line 259.

  - In Table 1, for the AntMaze Large-Div task, Diff-QL achieves the highest mean score but is not highlighted in bold.

---

> ### Author Rebuttal · Authors · 2026-03-30
>
> We sincerely thank the reviewer for recognizing the importance of the problem and the intuitive design of our three-stage pipeline. We address the concerns and questions below.
>
> **W1: Direct Evidence for Support Preservation:**
> We agree that stronger direct evidence of support preservation would further strengthen the paper. In the revision, we will add a compact dataset-reference support diagnostic with both quantitative and visual evidence, as shown in https://anonymous.4open.science/r/ICML2026-31415-Rebuttal-277D/fig1.png. We compare dataset actions with actions generated by SPAR-PROJ and SPAR-PLAS. Quantitatively, we will report a kNN-based support-distance metric in the original action space; lower values indicate closer alignment with the empirical behavior manifold. Visually, we will provide a concise shared-space boundary plot. Together, these diagnostics more directly test whether SPAR-PROJ stays closer to dataset support than SPAR-PLAS.
>
> **W2: Ablation on the Three Key Components:**
> Thank you for this suggestion. While the current paper includes related ablations, we agree the three main components should be evaluated more directly. In the revision, we will consolidate them into one component ablation table: (a) self-imitation in global space, (b) residual learning without latent self-imitation, and (c) removing Stage III gating, as shown in [T1]:
>
> |Task|cvae-base|cvae-w/-self-imitation(a)|res-w/o-latent-SI(b)|w/o-Stage-III(c)|SPAR-full|
> |:-:|:-:|:-:|:-:|:-:|:-:|
> |hp-mr|28.2|12.1|72.7|49.5|101.9|
> |hc-me|60.3|1.6|94.1|96.1|97.0|
> |pen-cl|58.8|3.9|67.8|61.6|76.2|
> |am-ld|15.0|0.0|40.0|60|70.0|
>
> The results show that all three components are necessary, and the full pipeline performs best overall, especially on the harder tasks.
>
> **W3&Minor Issues: Recent Baselines and Presentation Corrections:**
> We thank the reviewer for pointing out DAC, FQL, and QIPO. We will expand the related-work discussion and add comparisons to these recent methods, as shown in [T2]:
>
> |Env|SPAR|DAC|FQL|QIPO|
> |:-:|:-:|:-:|:-:|:-:|
> |hp-mr|101.9|99.8|64.8|101.2|
> |hc-me|97.0|47.6|106.1|94.0|
> |pen-cl|76.2|78.7|52.7|35.0|
> |am-ld|70.0|60.0|40.0|40.0|
>
>
>  We will also fix the reference display issues in Lines 46 and 259, and correct the boldface in Table 1 of the paper for the AntMaze Large-Div result.
>
> **Q1: Why is the weight applied only to the reconstruction term in the weighted ELBO:**
> We apply the advantage-based weight only to the reconstruction term because its purpose is to emphasize fitting high-value residual actions, whereas the KL term serves as a global latent regularizer. If the same weight were also applied to the KL term, high-advantage samples could be overly regularized toward the prior, which would weaken the model’s ability to preserve useful multimodal residual structure. Weighting only the reconstruction term therefore achieves the intended value-aware fitting while keeping latent regularization stable.
>
> **Q2: Why use CVAE instead of a more expressive generative model such as diffusion:**
> We choose CVAE because SPAR only needs to model the local residual distribution around a frozen BC anchor, rather than the full action distribution from scratch. This is a substantially easier modeling problem, for which CVAE provides a favorable efficiency-performance trade-off. We also tested stronger generative policies in the same residual-learning framework. [T3] compares MLP/flow bases and flow residual variants under matched settings. The main finding is that more expressive generators do not automatically yield better final performance, while SPAR remains strongest overall.
>
> |Task|mlp base|mlp base with flow policy|flow base|flow base with flow policy|SPAR|
> |:-:|:-:|:-:|:-:|:-:|:-:|
> |hp-mr|42.9|97.5|31.4|96.2|101.9|
> |hc-me|63.1|75.5|69.3|77.8|97.0|
> |pen-cl|50.4|68.2|64.6|68.5|76.2|
> |am-ld|20.0|5.0|10.0|10.0|70.0|
>
> We will clarify that our choice of CVAE is motivated by the localized nature of the residual space and the empirical trade-off we observed, rather than by any claim of universal superiority over diffusion- or flow-based generators.
>
> **Q3: Does a simple base policy limit the effectiveness of residual learning on multimodal datasets:**
> We clarify that, in SPAR, the base policy is intended to be a stable anchor, not a complete solution: it captures coarse behavior support, while the residual policy models local corrections, including multimodal residual structure in SPAR-PROJ. As shown in [T3] above, a stronger base policy does not necessarily yield a stronger final performance. We also evaluate SPAR residual learning on top of a flow-based base policy in [T4]. The result shows that SPAR can still substantially improve suboptimal but diverse bases, supporting that the framework is not limited to simple BC anchors.
>
> |Task|Flow-base Score|Residual Policy|SPAR Score|
> |:-:|:-:|:-:|:-:|
> |hp-mr|31.4|SPAR-MLP|76.5|
> |hc-me|69.3|SPAR-PROJ|94.9|
> |pen-cl|64.6|SPAR-PROJ|65.3|
> |am-ld|10.0|SPAR-PROJ|40.0|

---

> > ### Author Rebuttal · Reviewer_i7xa · 2026-04-06
> >
> > I appreciate the detailed responses. Consequently, I increased the score.

---

> > > ### Author Response · Authors · 2026-04-07
> > >
> > > Thank you for confirming resolution. We will incorporate all discussed additions in the rivision version.

---

### Official Review · Reviewer_pa4B · 2026-03-13

**Soundness:** 2
**Presentation:** 2
**Significance:** 2
**Originality:** 2
**Overall Recommendation:** 4
**Confidence:** 4

**Summary:**

This paper proposes an innovative three-stage offline strategy, whose core idea lies in decomposing global optimization into local residual correction around baseline anchors and mitigating the gradient conflict between fitting and improvement through latent variable self-imitation. The strategy demonstrates considerable novelty and promising results both theoretically and empirically, particularly exhibiting competitive performance in multimodal residual tasks and high-dimensional scenarios.

**Compliance With Llm Reviewing Policy:**

Affirmed.

**Final Justification:**

I think my concerns are addressed. I believe the work deserves to be accepted to ICML. I have increased the score to 4.

**Key Questions For Authors:**

Could you provide specific metrics(wall time...) for inference latency and memory consumption? Can you conduct a detailed analysis of the hyperparameter sensitivity? Additionally, would it be possible to evaluate the algorithm's performance on the OGbench benchmark?

**Limitations:**

yes

**Strengths And Weaknesses:**

**Strengths**: By decomposing global optimization into local residual correction, SPAR enables efficient improvement while preserving the original data distribution, thereby reducing the risk of distribution drift caused by the pursuit of high rewards.

**Weakness**: The algorithm requires tuning of numerous hyperparameters, features a relatively complex implementation structure, and incurs substantial overhead during the inference phase. Additionally, it exhibits high sensitivity to the quality of the baseline policy $π_{base}$ and the discriminative network $Q_{rob}$. Specifically, the thresholds $η_{abs}$, $η_{rel}$, $K$, and $T$ in Stage III all need to be manually configured, and the paper presents multiple "task-dependent" settings (e.g., distinct strategies for tasks such as AntMaze and Pen). Furthermore, relying solely on the D4RL dataset raises legitimate concerns about potential overfitting.

---

> ### Author Rebuttal · Authors · 2026-03-30
>
> We sincerely thank the reviewer for recognizing the novelty and potential improvements of our work. We address the concerns and questions below.
>
> **W1&Q2: Tuning Hyperparameters:**
> We clarify that the key hyperparameters in Stages I and II ($\lambda_u, \lambda_g, T$)have already been studied in Sec. 4.4. Specifically, Tables 3, 5, and 6 in the paper show predictable trends and stable performance across diverse tasks, supporting that these parameters are not brittle. The discussion of hyperparameters in stage III is in W5, which also shows the robustness.
>
> **W2: Implementation Complexity:**
> We respectfully note that the 3-stage design is intended to decouple optimization rather than complicate it. In particular, SPAR separately trains a standard BC policy, a Q estimator, and a lightweight residual gate. This avoids the instability of coupled actor-critic or min-max objectives and leads to much more reliable empirical optimization. We will make this design motivation more explicit in the revision.
>
> **W3&Q1: Inference Overhead:**
> We clarify that SPAR performs action generation in $O(1)$ time, unlike diffusion-based baselines that require 15–100 iterative denoising steps. Stage III only evaluates a small candidate set($K=10$). To quantify this, [T1] reports average inference latency and peak memory usage, measured with batch size 1 and averaged over 1,500 inference steps. The result is consistent: SPAR-PROJ is both faster and more memory-efficient than DAC.
>
> |Env|SPAR-PROJ(ms)|SPAR-PROJ(MB)|DAC(ms)|DAC(MB)|
> |:-:|:-:|:-:|:-:|:-:|
> |pen-cl|0.394|1412.52|0.577|1608.66|
> |hp-mr|0.366|1389.48|0.589|1590.12|
> |hc-me|0.409|1402.66|0.616|1592.66|
> |am-ld|0.403|1396.74|0.535|1592.79|
>
> **W4: Sensitivity to Baseline $\pi_{base}$ and Critic $Q_{rob}:$**
> - **Robustness to Suboptimal $\pi_{base}$:** We clarify that SPAR is not highly sensitive to baseline policy quality. As a residual rectification framework on top of a fixed backbone, it does not rely on a particular baseline family or a narrowly tuned starting point. Table 1 in the paper already shows consistent gains over BC baselines with different capacities, and [T2] further confirms clear improvements on a flow-matching baseline with a different inductive bias：
>
> |Task|Flow-base Score|Residual Policy|SPAR Score|
> |:-:|:-:|:-:|:-:|
> |hp-mr|31.4|SPAR-MLP|76.5|
> |hc-me|69.3|SPAR-PROJ|94.9|
> |pen-cl|64.6|SPAR-PROJ|65.3|
> |am-ld|10.0|SPAR-PROJ|40.0|
>
>
> - **Stable Critic Estimation ($Q_{rob}$):**  For the critic, we use in-sample expectile regression with $τ=0.5$, which recovers the SARSA-style conditional mean and avoids the OOD maximization that is a major source of overestimation. In addition, an ensemble lower-confidence-bound (LCB) penalty suppresses high-variance actions. Together, these two mechanisms make critic estimates conservative and stable, reducing sensitivity to imperfect initial modules.
>
> **W5: Stage III Thresholds:**
> We emphasize that the Stage III sensitivity analysis in this rebuttal focuses on the two main thresholds used for value gating and we use a shared configuration, ($\eta_{abs}=10^{-4}, \eta_{rel}=0.01$) across tasks, so Stage III acts as a conservative safety filter rather than a task-specific performance knob. This avoids giving SPAR extra task-specific tuning advantages relative to baselines. We add an ablation experiment on the stage III thresholds in [T3]:
>
> |$\eta_{abs}$|$\eta_{rel}$|pen-cl|hp-mr|hc-me|am-ld|
> |:-:|:-:|:-:|:-:|:-:|:-:|
> |1e9|1e9|50.4|42.9|63.1|20.0|
> |-1e9|-1e9|61.6|49.5|96.1|60.0|
> |-1e9|0.01|68.1|95.2|92.0|55.0|
> |1e-4|-1e9|67.2|78.2|96.4|55.0|
> |1e-4|0.01|76.2|101.9|97.0|70.0|
>
> **W6: Task-Dependent Settings:** We clarify that constructing different architectures aims to comprehensively reveal their respective applicability. For practice, SPAR-PROJ can be adopted as a unified choice. It adapts well to multimodal distributions, whereas deterministic policies suffer from mean-seeking behavior.
>
> **W7: Overfitting Concerns on D4RL:**
> We respectfully argue that SPAR is intrinsically less prone to overfitting than diffusion-based global methods. Under a comparable parameter budget, SPAR has a smaller effective hypothesis space because improvement is anchored to a frozen BC backbone and restricted to local residual correction, rather than unconstrained global generation. Moreover, the latent self-imitation mechanism keeps updates within value-weighted residual support, which helps reduce off-support drift and suppress critic-induced artifacts. This is why SPAR is both conceptually and empirically less sensitive to overfitting.
>
> **Q3: Evaluation on OGBench:**
> We deeply appreciate the suggestion. Given the limited rebuttal window, we will prioritize the requested latency/memory metrics and Stage III ablations, which most directly address your core concerns. We will strive to include OGBench results in the revision if computational resources permit.

---

> > ### Author Rebuttal · Reviewer_pa4B · 2026-04-03
> >
> > Thanks for your reply. Some of my concerns are addressed. However, the rebuttal results fail to help me understand why in Table 1 only one of  SPAR-mlp or SPAR-proj is used. I still find the hyperparameters need to be tuned with hard work. So currently I choose to keep my score considering the practical difficulty of the proposed method.

---

> > > ### Author Response · Authors · 2026-04-07
> > >
> > > Dear Reviewer pa4B,
> > >
> > > Thank you for your follow-up and your careful consideration of our work. We greatly appreciate your constructive insights, and we would like to address your remaining concerns in detail.
> > >
> > > ### 1. Table Merging and Reported Results
> > >
> > > To ensure clarity, we will merge the results for both SPAR-MLP and SPAR-PROJ into a single, comprehensive Table 1 in the revision so that both variants are visible side by side for every task. We initially reported one representative variant per domain to keep the table concise, but we fully agree that showing both variants together provides a more transparent and informative comparison.
> > >
> > > We would also like to highlight that, for practical deployment, SPAR-PROJ can serve as a unified, architecture-agnostic default: its generative parameterization seamlessly accommodates both unimodal and multimodal residual distributions without requiring users to choose between projection heads. This means that a practitioner can simply adopt SPAR-PROJ across all tasks and still obtain competitive or superior performance, which we believe significantly lowers the barrier to real-world adoption.
> > >
> > > ### 2. Hyperparameter Sensitivity and Tuning Effort
> > >
> > > We appreciate this concern and would like to clarify that SPAR's tuning burden is, in fact, lighter than that of several prominent baselines. SPAR has two main task-specific hyperparameters $(\lambda_g, \lambda_u)$ . As shown in Tables 3 and 6, both exhibit stable and interpretable trends across tasks, and a single default configuration already works well on 8 out of 10 tasks without any task-specific adjustment. Stage III further uses a fixed shared threshold across all tasks, requiring no per-task calibration.
> > >
> > > Crucially, the decoupled three-stage design is a key structural advantage: each hyperparameter only affects its designated training stage, eliminating the need for joint tuning. Practitioners can adjust each knob independently with predictable, isolated effects on performance. This stands in contrast to methods where multiple hyperparameters interact across a single training loop.
> > >
> > > To put this in perspective, task-specific hyperparameter tuning is standard practice among state-of-the-art offline RL methods:
> > >
> > > - Diffusion-QL [1] requires tuning the guidance weight $η$  per task to balance the Q-value gradient against the behavior-cloning diffusion loss, and its performance is known to be sensitive to this trade-off.
> > > - DAC [2] requires tuning both the LCB penalty coefficient and the KL regularization strength $η$, both of which interact with the actor-critic training dynamics.
> > > - qipo [3] relies on a temperature parameter $β$ that controls the sharpness of the energy weighting, which directly determines how aggressively the policy deviates from the behavior distribution and needs task-specific calibration.
> > >
> > > In all three cases, the hyperparameters interact with the critic or energy model during a single coupled training phase, making their effects harder to isolate and predict. By contrast, SPAR's three-stage decoupling ensures that tuning Stage I does not affect Stage II, and vice versa. We believe this structural simplicity is a meaningful practical advantage that is often overlooked when comparing raw hyperparameter counts.
> > >
> > > ### 3. Stage III Design and Task-Specific Thresholds
> > >
> > > We would like to further emphasize the design philosophy behind Stage III. The fixed shared threshold serves as a lightweight safety filter that clips extreme actions without introducing additional task-specific complexity. This is deliberately minimal: Stage I already provides robust, task-independent baseline optimization through stable residual correction, and Stage II handles the guidance signal.
> > >
> > > Stage III therefore only needs to ensure that the final output remains within a reasonable bound, which can be achieved with a single universal threshold. This contrasts with methods that require manually configured, task-dependent clipping or penalty schedules, adding another layer of tuning overhead.
> > >
> > > Once again, thank you for your thoughtful and rigorous review. Your feedback has been invaluable in helping us improve the clarity of our presentation. We will incorporate all discussed clarifications and table restructuring into the camera-ready version, and we are confident that the revised manuscript will address all points raised. We look forward to presenting the updated version for your further evaluation.
> > >
> > > [1] Wang Z, Hunt J J, Zhou M. *Diffusion policies as an expressive policy class for offline reinforcement learning*. arXiv preprint arXiv:2208.06193, 2022.
> > >
> > > [2] Fang L, Liu R, Zhang J, et al. *Diffusion Actor-Critic: Formulating Constrained Policy Iteration as Diffusion Noise Regression for Offline Reinforcement Learning*. International Conference on Learning Representations, 2025.
> > >
> > > [3] Zhang S, Zhang W, Gu Q. *Energy-Weighted Flow Matching for Offline Reinforcement Learning*. International Conference on Learning Representations, 2025.

---

### Decision · Program_Chairs · 2026-04-30

**Decision:**

Accept (regular)

**Comment:**

This paper introduces SPAR, which reframes offline policy improvement as learning corrective adjustments around a fixed behavior cloning baseline. By separating behavior modeling from value-guided refinement, the method offers a principled mechanism to keep optimization within the data distribution while still achieving meaningful policy gains. This residual-policy formulation represents a conceptually fresh perspective.

The proposed latent self-imitation (LSI) replaces traditional gradient-based policy updates with value-weighted sampling over a learned residual space. This serves as an intriguing alternative to conventional gradient ascent in offline RL, specifically designed to reduce policy drift beyond the behavior support stemming from critic estimation errors.

The authors responded thoroughly to all feedback and provided additional experimental results. Every review concern has been adequately addressed.